# EFFICIENT HYPERPARAMETER TUNING VIA TRAJECTORY INVARIANCE PRINCIPLE

## ABSTRACT

As hyperparameter tuning becomes increasingly costly at scale, efficient tuning methods are essential. Yet principles for guiding hyperparameter tuning remain limited. In this work, we seek to establish such principles by considering a broad range of hyperparameters, including batch size, learning rate, and weight decay. We identify a phenomenon we call *trajectory invariance*, where pre-training loss curves, gradient noise, and gradient norm exhibit invariance–closely overlapping– with respect to a quantity that combines learning rate and weight decay. This phenomenon effectively reduces the original two-dimensional hyperparameter space to one dimension, yielding an efficient tuning rule: follow the salient direction revealed by trajectory invariance. Furthermore, we refine previous scaling laws and challenge several existing viewpoints. Overall, our work proposes new principles for efficient tuning and inspires future research on scaling laws.

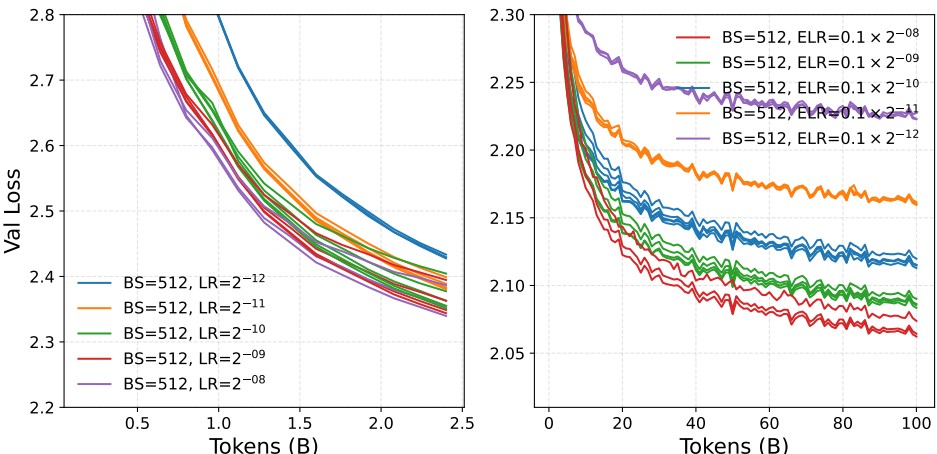

Figure 1: **Trajectory invariance and its shift during training. (a) Invariance w.r.t. learning rate (LR): Curves with the same LR overlap initially and start to diverge.** Validation loss curves for different LRs $\eta$ (different colors) and varying weight decays (WDs) $\lambda$ under the same LR $\eta$ (same color). For example, all blues curves correspond to $(\eta, \lambda) \in \{(2^{-12}, 0.4), (2^{-12}, 0.2), (2^{-12}, 0.1)\}$, where LR is fixed at $2^{-12}$. **(b) Invariance w.r.t. effective learning rate (ELR): Curves with the same ELR stably overlap up to 100B tokens.** Loss curves for different ELRs ($\gamma := \eta\lambda$) (different colors) and varying $(\eta, \lambda)$ under the same ELR (same color). For example, all blues curves correspond to $(\eta, \lambda) \in \{(2^{-12}, 0.4), (2^{-11}, 0.2), (2^{-10}, 0.1), (2^{-9}, 0.05), (2^{-8}, 0.025)\}$, where ELR is fixed at 1e-4.

## 1 INTRODUCTION

Hyperparameter tuning plays a critical role in deep learning, as it directly influences both training stability and final model performance (Kaplan et al., 2020; Hoffmann et al., 2022; Wortsman et al., 2024). However, grid search is becoming prohibitively expensive: sweeping learning rate and batch

size over a $5 \times 5$ grid with a 3B model over 30B tokens demands roughly 1e22 FLOPs. It is urgent to develop principles to guide efficient hyperparameter tuning.

Prior work has investigated principles of hyperparameter tuning, but the scope has often been narrow. Many studies concentrate on a limited subset of hyperparameters–such as batch size and learning rate—while overlooking others like weight decay (e.g., DeepSeek-AI (2024); Porian et al. (2024)). In addition, experimental setups often differ in architecture choices or learning-rate schedules. As a result, there are many divergent or even contradicted principles in the field. For example, regarding learning rate, Li et al. (2025) argue it should increase with data size, Bjorck et al. (2025) suggest it should decrease, while Bergsma et al. (2025a) recommend leaving it unchanged. Similarly, for batch size, Li et al. (2025); Bergsma et al. (2025a) advocate scaling it up with data, whereas Marek et al. (2025) recommend to avoid gradient accumulation. Collectively, these discrepancies highlight that existing principles for hyperparameter tuning are insufficient, and our understanding of hyperparameter scaling in large-scale training remains limited. In this paper, we seek to find principles within a more rigorous experimental setup that considers a broader range of hyperparameters, including batch size, learning rate, weight decay, and data size.

We identify a surprising phenomenon which we call *trajectory invariance*: the pre-training loss curves exhibit invariance—closely overlap–w.r.t. a quantity that combines learning rate $\eta$ and weight decay $\lambda$. Specifically, in the early training stage, this invariant quantity is simply the learning rate itself (Figure 1 (a)). As training progresses, the quantity becomes the product of learning rate and weight decay, i.e., the effective learning rate (Figure 1 (b)). The late-stage invariance w.r.t effective learning rate is particularly striking: for example, for two runs with $(\eta, \lambda) \in \{(2^{-11}, 0.4), (2^{-8}, 0.05)\}$, despite the learning rates differ by a factor of 8, their loss curves surprisingly overlap (orange curves in Figure 1(b)).

The trajectory invariance phenomenon provides a practical principle for efficient hyperparameter tuning. By following the salient direction in the hyperparameter space revealed by trajectory invariance (e.g. ELR), we can effectively reduce the dimension of hyperparameter tuning space (e.g. fixing LR and only tuning weight decay).

Our results also challenge multiple existing hyperparameter tuning principles based on limited hyperparameter space. For example, we find that scaling law for LR may be limited in its current scope, and the scaling law for optimal BS may not fully generalize to very large BS. Therefore, to get general principles for hyperparameter tuning like trajectory invariance, we call for future work to study in a larger hyperparameter space.

Overall, our work proposes the trajectory invariance principle and refines hyperparameter scaling principles, offering new practical guidelines for efficient hyperparameter tuning. The trajectory invariance phenomenon leads us to rethink hidden structure of the hyperparameter space and scaling laws research. We expect that richer structures exist in higher-dimensional hyperparameter spaces when factors such as initialization $\sigma$ and Adam's $\beta_1$, $\beta_2$ are included. Identifying such invariance structures can make hyperparameter tuning more efficient by reducing the search space, covering more of the hyperparameter space with fewer resources. Eventually, powerful efficient tuning methods may help us to explore the maximal benefits and ultimate limits of hyperparameter tuning.

## 2 PRELIMINARIES

**Experiment setup.** We train a series of autoregressive language models with 164M[1] parameters. For the model architecture, we use SwiGLU MLP (Shazeer, 2020), rotary positional embeddings (Su et al., 2024), RMSNorm, and untied embedding parameters. We adopt the GPT-2 tokenizer and train our models on the Pile dataset (Gao et al., 2020), with up to 100B tokens (about 600 tokens-per-parameter, TPP). We use the AdamW optimizer (Kingma & Ba, 2015; Loshchilov & Hutter, 2017) with default hyperparameters $\beta_1 = 0.9$, $\beta_2 = 0.95$, and gradient clipping of $1.0$.

Our experiments tune batch size (BS), learning rate (LR), and weight decay (WD), and study their effects. We mainly use two BS values, $B \in \{512, 8192\}$, in unit of sequence, with a context length of 1024. We conduct two-dim sweeps varying LR $\eta$ over the grid $\{2^{-12}, 2^{-11}, 2^{-10}, 2^{-9}, 2^{-8}\}$ and WD $\lambda$ over the grid $\{0.025, 0.05, 0.1, 0.2, 0.4\}$. We study both the constant learning rate scheduler

---

[1]The scale allows for richer and more diverse experimental designs through hyperparameter sweeps.

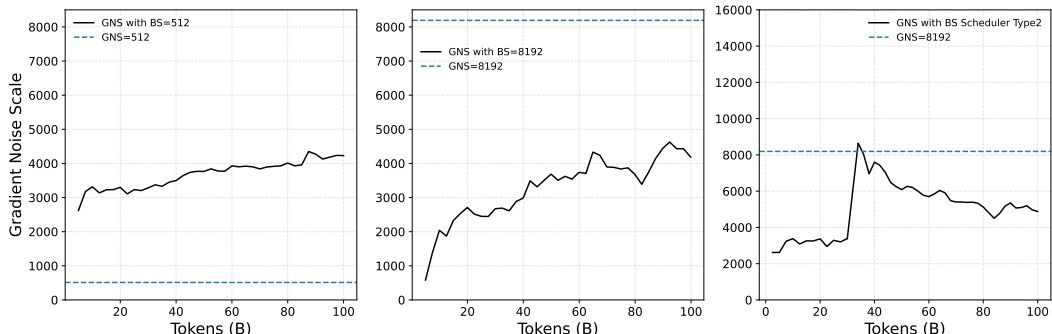

Figure 2: **Gradient noise scale** $\mathcal{B}_{\text{precond}}$ **dynamics. (a) Dynamics with** $B = 512$**.** The dotted line (GNS = 512) equals the batch size actually used. Therefore, $B = 512$ is a small BS, since $\mathcal{B}_{\text{precond}}$ remains larger than $512$ throughout training. **(b) Dynamics with** $B = 8192$**.** $B = 8192$ is a large BS, as $\mathcal{B}_{\text{precond}}$ stays smaller than $B$. **(c) Dynamics with a batch size scheduler.** BS Scheduler effectively increases $\mathcal{B}_{\text{precond}}$, allowing for larger batch sizes. See GNS for more runs in Figure 28.

and the WSD (Hu et al., 2024) scheduler, always using 1B tokens for warm-up in both cases. In the decay stage, we linearly decay the learning rate to zero over 5B tokens.

**Theoretical notation.** We denote batch size, learning rate, and weight decay, effective learning rate by $B$, $\eta$, $\lambda$, and $\gamma$, respectively. Denote the $\ell^2$ norm by $\|\cdot\|$. For two sequences $\mathbf{a}_1, \mathbf{a}_2$, the relative distance $\|\cdot\|_{\text{rel}}$ is defined as $\|\mathbf{a}_1 - \mathbf{a}_2\|_{\text{rel}} := \|\mathbf{a}_1 - \mathbf{a}_2\| / \sqrt{1/2(\|\mathbf{a}_1\|^2 + \|\mathbf{a}_2\|^2)}$.

Next, we introduce interested quantities in our paper. Let $\mathcal{D}$ denote the data distribution, $\mathbf{w}$ the model parameters, and $L(\mathbf{w}, \mathbf{x})$ the loss for one sequence $\mathbf{x} \sim \mathcal{D}$. Define the stochastic gradient and the population gradient $\tilde{\mathbf{g}}_{\mathbf{x}} := \nabla_{\mathbf{w}} L(\mathbf{w}, \mathbf{x})$, $\mathbf{g} := \mathbb{E}[\nabla_{\mathbf{w}} L(\mathbf{w}, \mathbf{x})]$. We then have $\mathbf{g} = \mathbb{E}[\tilde{\mathbf{g}}_{\mathbf{x}}]$. The *population gradient norm square* is $\|\mathbf{g}\|^2$. Let the gradient noise covariance be $\boldsymbol{\Sigma} := \mathbb{E}[(\tilde{\mathbf{g}}_{\mathbf{x}} - \mathbf{g})(\tilde{\mathbf{g}}_{\mathbf{x}} - \mathbf{g})^\top]$. We primarily care about its trace, $\text{Tr}(\boldsymbol{\Sigma})$, which we call *gradient noise*. The corresponding *gradient noise scale* (GNS) (McCandlish et al., 2018) is defined as $\mathcal{B}_{\text{simple}} := \text{Tr}(\boldsymbol{\Sigma}) / \|\mathbf{g}\|^2$.

For Adam, define the diagonal preconditioner $\mathbf{P} := \text{diag}(\mathbf{v}^{1/2})$, so that Adam can be viewed as a preconditioned method with $\mathbf{P}^{-1}$. We then define *preconditioned* version of gradient noise, population gradient norm square, and GNS as $\text{Tr}(\mathbf{P}^{-1}\boldsymbol{\Sigma})$, $\|\mathbf{P}^{-1/2}\mathbf{g}\|^2$, and $\mathcal{B}_{\text{precond}} := \text{Tr}(\mathbf{P}^{-1}\boldsymbol{\Sigma}) / \|\mathbf{P}^{-1/2}\mathbf{g}\|^2$, respectively.

**Effective learning rate (ELR).** It is well understood that the learning rate alone does not faithfully represent the speed of learning. To address this, the notion of *effective learning rate* has been introduced to better capture this intuition. Various definitions of ELR exist; here, we adopt the simplest form—defined as the product of the learning rate and weight decay, i.e., $\gamma := \eta\lambda$ (Zhang et al., 2019; Li & Arora, 2019; Wan et al., 2021; Bergsma et al., 2025a). Zuo et al. (2025) also employ the $\sqrt{\eta\lambda}$ formulation. Other works interpret ELR from different perspectives, such as viewing AdamW as sign gradient descent (D'Angelo et al., 2024) or analyzing the expected rotational update size (Kosson et al., 2024).

Specifically, the ELR model, inspired by SignGD (D'Angelo et al., 2024), and parameter-norm analysis (Kosson et al., 2024) provide a useful link between different definitions of ELR. D'Angelo et al. (2024) define ELR as $\eta / \|\mathbf{w}\|$ to capture the actual step size, while Kosson et al. (2024) show that the parameter norm stabilizes during training as $\|\mathbf{w}\| \propto \sqrt{\eta/\lambda}$. Once the parameter norm stabilizes, the resulting ELR reduces to $\sqrt{\eta\lambda}$. This matches our definition of ELR and reflects the intuition of training speed.

**Identifying small and large batch sizes with gradient noise scale (GNS).** McCandlish et al. (2018) introduced the notion of *gradient noise scale*, $\mathcal{B}_{\text{simple}}$, which marks the critical point where the LR-BS linear scaling rule of SGD breaks, which states that when scaling $B \rightarrow kB$, the LR can be scale as $\eta \rightarrow k\eta$ to maintain similar training dynamics. It also serves as an estimate of the *critical batch size* (CBS), beyond which increasing $B$ yields diminishing returns. Thus, a larger $\mathcal{B}_{\text{simple}}$ permits a larger $B$ without performance loss.

In this work, we use the preconditioned GNS, $\mathcal{B}_{\text{precond}}$, as an estimate of CBS for AdamW. We classify $B$ as a *small batch size* if $\mathcal{B}_{\text{precond}} > B$ throughout training, and as a *large batch size* if $\mathcal{B}_{\text{precond}} < B$. As shown in Figure 2, $B = 512$ is a small batch (Figure 2(a)), while $B = 8192$ is a large batch (Figure 2(b)).

## 3 TRAJECTORY INVARIANCE PRINCIPLE

In this section, we identify the phenomenon of *trajectory invariance* and discuss its implications. Specifically, early in training, loss curves overlap when the learning rate is fixed and weight decay varies, showing invariance with respect to learning rate. As training progresses, this invariance shifts to the effective learning rate. We find that the emergence of trajectory invariance w.r.t. ELR is more related to the number of training iterations than to the total number of tokens processed. Finally, trajectory invariance leads to a new principle for efficient tuning, reducing the two-dimensional hyperparameter space of learning rate and weight decay to a single tuning direction.

### 3.1 TRAJECTORY INVARIANCE WITH RESPECT TO LR AND ELR

We conduct experiments to systematically investigate trajectory invariance phenomenon. Specifically, we train language models over a two-dimensional grid of LR $\eta \in \{2^{-12}, 2^{-11}, 2^{-10}, 2^{-9}, 2^{-8}\}$ and WD $\lambda \in \{0.025, 0.05, 0.1, 0.2, 0.4\}$ at a fixed BS of 512 (0.5M tokens). This effectively leads to a sweep of ELR $\gamma \in \{6.25\text{e-}6, 1.25\text{e-}5, \ldots, 8\text{e-}4, 1.6\text{e-}3\}$ (note that $2^{-10} \approx 1\text{e-}3$).

**Trajectory invariance of loss curves w.r.t. LR at early training.** We begin by examining trajectory invariance w.r.t. LR. In Figure 1 (a), validation loss curves are shown for different LRs (different colors) and varying WDs under the same LR (same color). We focus on the first 5B tokens–about 30 TPP, close to the Chinchilla-optimal setting of 20 TPP. At this stage, loss curves overlap when LR is fixed and WD varies. For example, with $(\eta, \lambda) \in \{(2^{-12}, 0.4), (2^{-12}, 0.2), (2^{-12}, 0.1)\}$, the LR remains constant at $2^{-12}$, and all corresponding runs converge to nearly identical trajectories (blue curves in Figure 1 (a)). Moreover, invariance with respect to LR persists longer for runs with smaller LRs. Over time, runs with the same LR diverge and transition into the next stage: trajectory invariance w.r.t. ELR.

**Trajectory invariance w.r.t. ELR with sufficient training.** We next examine trajectory invariance w.r.t. ELR. Figure 1 (b) shows validation loss curves for different ELRs (different colors), along with curves for different combinations of LR and WD that yield the same ELR (same colors). When LR and WD vary but ELR is fixed, the loss curves overlap closely. For example, with $(\eta, \lambda) \in \{(2^{-12}, 0.4), (2^{-11}, 0.2), (2^{-10}, 0.1), (2^{-9}, 0.05), (2^{-8}, 0.025)\}$, the ELR remains constant at 1e-4, and all corresponding runs converge to nearly identical curves (blue curves in Figure 1 (b)). In contrast, curves with different ELRs separate into distinct clusters, with differences across clusters being much larger than within clusters. Finally, the strength of invariance itself depends on ELR: smaller ELRs yield slightly weaker alignment.

To quantify trajectory invariance, we compute the relative distance $\| \cdot \|_{\text{rel}}$ between loss curves. Over a wider range of ELRs than in Figure 1 (b), we plot the pairwise relative distances between all runs in Figure 3 (a), and the averaged distance between clusters in Figure 3 (b). Both analyses provide strong evidence for trajectory invariance with respect to ELR.

Additionally, this phenomenon extends beyond validation loss. Key optimization metrics such as gradient noise and gradient norm also exhibit trajectory invariance with respect to ELR (Figure 7 and Figure 10). Moreover, trajectory invariance persists when using the WSD scheduler (Figure 8), underscoring its robustness and universality.

**How trajectory invariance depends on ELR: A try with power law.** We try to dive deep about a precise characterization on the collapse: we already know loss (Figure 1) and gradient noise (Figure 7) are proportional to ELR, i.e., $L \propto \gamma$, $\text{Tr}(\mathbf{P}^{-1}\mathbf{\Sigma}) \propto \gamma$, but what is the exact relationship? We opt to fit the irreducible loss in the constant loss curves and stable gradient noise and find simple power laws can fit the trend well. Specifically, prior works (Tissue et al., 2024; Luo et al., 2025) find loss curves with constant LR $\eta$ and WD $\lambda$ can be well approximated with a power law:

$$L_{\eta,\lambda}(t) = L_{0,\eta,\lambda} + A_{\eta,\lambda} \cdot (\eta t)^{-\alpha_{\eta,\lambda}},$$

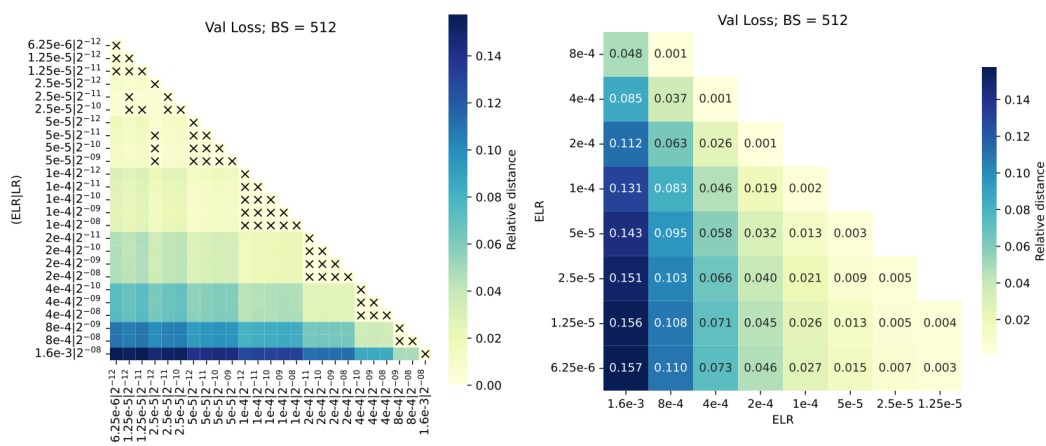

Figure 3: **Quantitative evaluation on trajectory invariance w.r.t. ELR based on relative distance** $\| \cdot \|_{\mathrm{rel}}$. **(a)** Pairwise relative distances between all runs. Small $\| \cdot \|_{\mathrm{rel}}$ indicates greater similarity between curves. A $\times$ marks distances below a threshold (0.005 here). **(b)** Averaged distance between groups defined by ELR. For example, the top-left point corresponds to the average $\|\mathbf{L}_1 - \mathbf{L}_2\|_{\mathrm{rel}}$ over all pairs of loss curves, where $\mathbf{L}_1$ has ELR $\gamma = 8\mathrm{e}\text{-}4$ and $\mathbf{L}_2$ has $\gamma = 1.6\mathrm{e}\text{-}3$.

where $t$ is number of iterations and $L_{0,\eta,\lambda}, A_{\eta,\lambda}, \alpha_{\eta,\lambda}$ are curve dependent parameters. We unify the expression $L_{0,\eta,\lambda}$ via the parameterization $L_{0,\eta,\lambda} = L_{0,1} + L_{0,2}(\eta\lambda)^{L_{0,3}}$. We leave the fitting on $A_{\eta,\lambda}$ and $\alpha_{\eta,\lambda}$ to future work. On the other hand, as shown in Figure 7 (b), gradient noise (denoted by $G_{\eta,\lambda}$ in this paragraph) increases very slowly and becomes stable as training proceeds, we therefore parameterize $G_{\eta,\lambda}$ as a constant over time: $G_{\eta,\lambda} = G_1(\eta\lambda)^{G_2}$.

Our fitting results are summarized in Table 1. Both $L_{0,\eta,\lambda}$ and $G_{\eta,\lambda}$ are well captured by the fits ($R^2 > 0.96$). The exponent for $L_{0,\eta,\lambda}$, $L_{0,3} \approx 0.46$, is close to $0.5$, which naturally recovers the form $\sqrt{\eta\lambda}$. The exponent for gradient noise, $G_2 \approx 0.36$, is–to the best of our knowledge–new in the literature and may be of independent interest. Predicted values across different $\gamma$ are shown in Fig. 9. Taken together, these results suggest that $(\eta, \lambda)$ has predictive power and may serve as a useful basis for describing loss curves across settings in a more universal form.

Table 1: **Fitting results of** $L_{0,\eta,\lambda} = L_{0,1} + L_{0,2}(\eta\lambda)^{L_{0,3}}$ **and** $G_{\eta,\lambda} = G_1(\eta\lambda)^{G_2}$**.**

| $L_{0,1}$ | $L_{0,2}$ | $L_{0,3}$ | $R^2$ | $G_1$ | $G_2$ | $R^2$ |
|---|---|---|---|---|---|---|
| 1.9585 | 9.2613 | 0.4604 | 0.9978 | 15.6582 | 0.3561 | 0.9601 |

## 3.2 SUFFICIENT ITERATIONS MATTERS FOR INVARIANCE WITH RESPECT TO ELR

Interestingly, we find that the emergence of trajectory invariance with respect to ELR is governed more by the number of training iterations than by the total number of tokens processed.

**Trajectory invariance stuck on LR at large BS.** We first justify this claim by decreasing training iterations with a larger BS. To this end, we repeat our two-dim sweep experiments with the same data size at a large BS of 8192 (8M tokens). As shown in Figure 4 (a), even with a 100B token budget, the large-batch setting exhibits trajectory invariance only with respect to the LR, resembling the pattern observed at small batch sizes in the limited-data regime (Figure 1 (a)). Notably, the number of iterations in these two settings—$B = 8192$ with 100B tokens (12,500 iterations) and $B = 512$ with 5B tokens (10,000 iterations)—is roughly comparable. This suggests that the number of training iterations, rather than the token budget, primarily drives the shift in trajectory invariance.

**Recovering trajectory invariance w.r.t. ELR with batch size schedulers.** We further validate our hypothesis by increasing the number of training iterations under large-batch settings. To recover

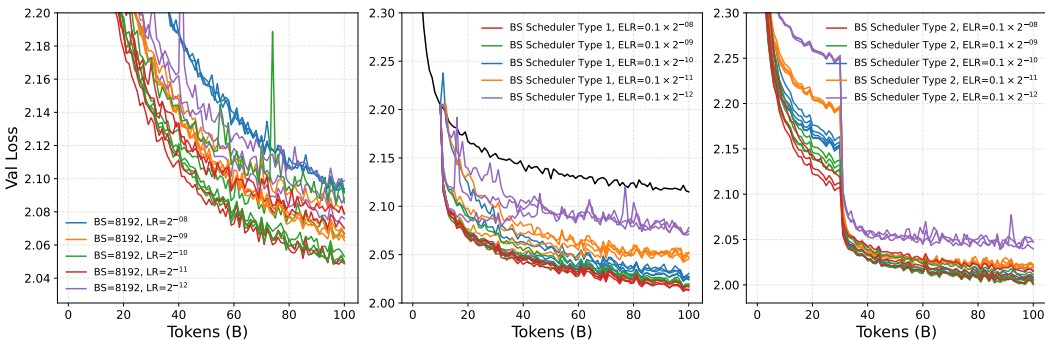

Figure 4: **Trajectory Invariance with large batch size and batch size schedulers. (a)** Large-batch training exhibits invariance only with respect to LR, even with sufficient tokens. The experiment settings match Fig. 1(b), except with a large batch size $B = 8192$. **(b) (c)** Two batch size schedulers successfully recover invariance with respect to ELR under large-batch training. Implementation details of batch size schedulers are provided in Sec. 3.2.

the trajectory invariance pattern with respect to the ELR while keeping the token budget fixed, we introduce a simple *batch size scheduler*: training begins with a small batch size ($B = 512$) and then switches to a large batch size ($B = 8192$) partway through. We consider two types of schedulers:

- *Type 1 (Fixed checkpoint)*: Resume large-batch training ($B = 8192$) from a small-batch checkpoint ($B = 512$), and sweep $\eta$ and $\lambda$ over a two-dimensional grid. The source checkpoint is trained on 10B tokens with $\eta = 2^{-10}$ and $\lambda = 0.1$.
- *Type 2 (Same source and target)*: Train with $B = 512$ from initialization while sweeping $\eta$ and $\lambda$, then switch to large-batch training ($B = 8192$) at a fixed point (30B tokens) without changing $\eta$ or $\lambda$.

As shown in Figure 4(b),(c), both BS schedulers successfully recover trajectory invariance w.r.t. ELR for loss curves, similar to small-batch training. This demonstrates that large batch size itself is not the determining factor for whether trajectory invariance w.r.t. ELR emerges; rather, it is the number of training iterations. Moreover, the presence of trajectory invariance across both schedulers highlights its robustness and generality. A final note is that gradient noise behaves differently under the two schedulers and we only discuss Type 2 scheduler subsequently. See Fig. 12 and Fig. 13 for details.

## 3.3 DISCUSSION WITH PRIOR WORK

**Two-stage behavior revealed by trajectory invariance.** The distinct patterns of trajectory invariance suggest a two-stage training process: In the early stage, the shape of the loss curves is determined mainly by LR, while as training proceeds, the loss values are governed primarily by ELR. The transition of invariance from LR to ELR thus serves as a marker for leaving the early stage of training. The speed of this transition is primarily governed by the number of iterations and the learning rate $\eta$. Prior work (Wang & Aitchison, 2025; Bergsma et al., 2025b) has shared the same viewpoint, but only at a conceptual level without the empirical validation presented here.

**Comparison with D'Angelo et al. (2024); Schaipp (2024).** Trajectory invariance with respect to ELR has been hinted at in prior work. Schaipp (2024) report that, in image classification tasks with multi-epoch training, final test accuracy remains nearly unchanged when varying learning rate and weight decay while keeping their product fixed (Figure 1 therein). In contrast, our study focuses on language model pretraining in a one-pass regime, and our observations are broader: we identify trajectory invariance not only for loss curves throughout training, but also for gradient noise and parameter norm.

D'Angelo et al. (2024) observe a similar phenomenon in image classification (Figures 3–4 therein). They distinguish between an "over-training regime" (multi-epoch classification) and an "under-training regime" (one-pass language modeling), and argue that weight decay operates through different mechanisms in each. However, our results (Figure 1) show that trajectory invariance also

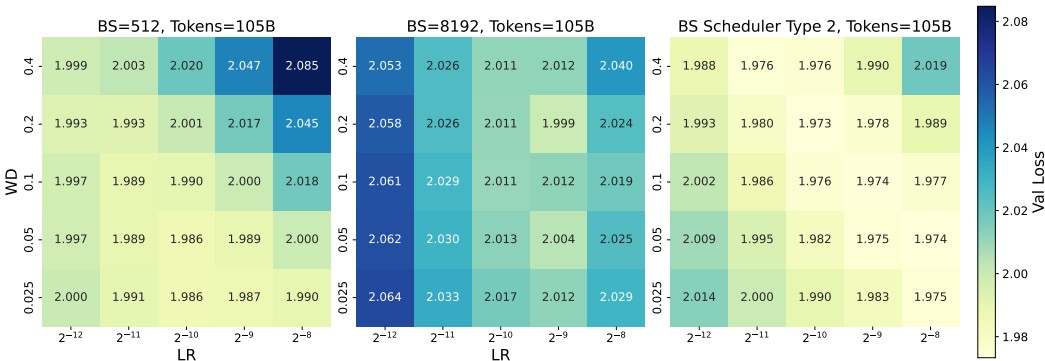

Figure 5: **Heatmap of final validation loss after training on 105B tokens with a WSD scheduler over a $5 \times 5$ grid of peak LRs and WDs.** **(a)** Small batch size ($B = 512$): ELR is the salient factor for final performance. **(b)** Large batch size ($B = 8192$): LR becomes the salient factor. **(c)** Batch size scheduler (Type 2): ELR becomes the salient factor again.

emerges in the one-pass regime given sufficient training, suggesting that the mechanisms of weight decay across regimes are not entirely disjoint. Moreover, we extend this line of inquiry by analyzing the role of batch size, finding that sufficient iterations are critical for trajectory invariance to appear.

### 3.4 EFFICIENT HYPERPARAMETER TUNING VIA TRAJECTORY INVARIANCE

The phenomenon of trajectory invariance reveals hidden structure in the hyperparameter space, which can be leveraged to design more efficient tuning strategies. By examining validation loss after LR decay, we demonstrate how trajectory invariance connects to final performance and motivates a practical guideline: tune along the salient direction indicated by invariance.

**Trajectory invariance connects to final performance.** To enable fair comparison of final performance, we report validation loss after LR decay using the WSD scheduler trained on 105B tokens (Figure 5). At small BS (Figure 5(a)), ELR is the dominant factor, and runs with the same ELR achieve similar performance. In contrast, at large BS (Figure 5(b)), raw LR becomes the main driver of performance, while WD only introduces minor variation. These results are fully consistent with the trajectory invariance patterns observed in the training loss curves (Figure 1(b) and Figure 4(a)).

We provide additional evidence for this trend by fitting quadratic functions (paraboloids) of loss with respect to LR and WD at various token horizons, and analyzing their spectra. In Figure 15, we present the dynamics of eigenvalues and top eigenvectors of the fitted paraboloids. Since the top eigenvalue consistently dominates, its corresponding eigenvector identifies the direction along which performance changes most significantly. For small batch sizes, these top eigenvectors align strongly with $(1, 1)$—the direction of varying ELR—while for large batch sizes, they align with $(1, 0)$—the direction of varying LR. This provides stronger confirmation of our earlier observations in Figures 5 (a) and (b).

**A simple tuning strategy.** To achieve efficient tuning with minimal resources, tuning should follow the salient direction revealed by trajectory invariance. For small batch sizes or in sufficient-iteration regimes, this means tuning along the direction that changes ELR. This contrasts with the recommendation in Schaipp (2024), which suggests keeping ELR fixed. For example, tuning either LR or WD individually is consistent with our guideline, and as we will show in the next section, fixing LR and tuning WD is preferable. On the other hand, for large batch sizes or in limited-iteration regimes, the recommended strategy is to tune LR while keeping WD fixed. This observation also explains why the common practice of fixing WD and tuning only LR works well in both settings.

## 4 REFINED HYPERPARAMETER SCALING LAWS ACROSS DATASET SIZE

In this section, we fit scaling laws for both hyperparameters and loss as functions of dataset size. For each token budget, we train with various peak LRs and WDs, decay the LR to zero with WSD

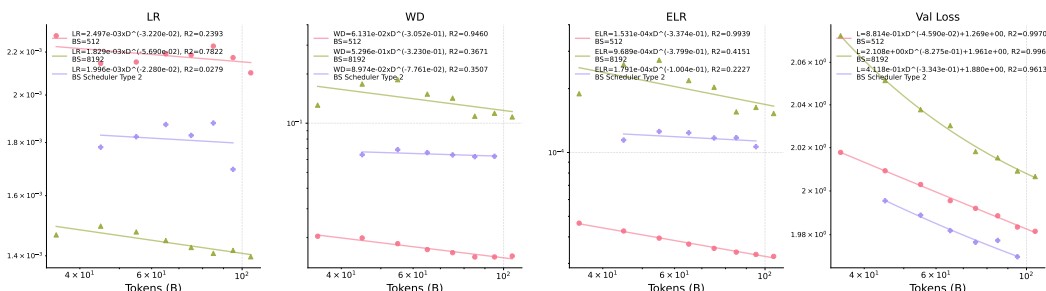

Figure 6: **Scaling laws for hyperparameters and loss across dataset size.** To identify optima at each token budget, we fit a paraboloid to the loss surface (as in Fig. 5). The minima of this paraboloid define the optimal LR and WD, while the corresponding loss value gives the optimal loss. **(a)** Optimal LR. **(b)** Optimal WD. **(c)** Optimal ELR, the product of optimal LR and WD. **(d)** Optimal loss.

scheduler, and then fit a paraboloid over the two-dim grid of LR and WD. The minimum of this fitted surface is taken as the optimal point for that budget. We then fit scaling laws using power functions and present the results in Figure 6. Based on our results, we discuss some existing claims and propose refined scaling laws.

$$\eta(D) = \eta_1 \cdot D^{\eta_2}, \quad \lambda(D) = \lambda_1 \cdot D^{\lambda_2}, \quad \gamma(D) = \gamma_1 \cdot D^{\gamma_2}, \quad L(D) = E + B \cdot D^{-\beta}.$$

**New scaling trends for optimal LR and WD.** Figure 6 illustrates the scaling behavior of optimal hyperparameters. As dataset size $D$ increases, the optimal WD $\lambda$ decreases sublinearly, while the optimal LR $\eta$ remains nearly constant (all exponents $|\eta_2| < 0.057$). Together, these trends imply that the optimal ELR $\gamma$ decreases with $D$, and the patterns are consistent across both small and large BSs. Moreover, these findings reinforce the simple tuning strategy proposed in Section 3.4: for small batch sizes or in sufficient-iterations regimes, it is preferable to fix $\eta$ and tune $\lambda$.

Our results on the scaling of LR and WD challenge and refine existing claims. The near-constant optimal LR contradicts prior suggestions that it should increase with data size (Li et al., 2025) or decrease (Bjorck et al., 2025; DeepSeek-AI, 2024), while supporting the view that LR need not be tuned (Bergsma et al., 2025a). Similarly, the sublinear decrease of optimal WD contrasts with the inverse-linear scaling proposed in (Wang & Aitchison, 2025), but is consistent with the findings of (Bergsma et al., 2025a).

**Challenging optimal batch size scaling law.** Recent studies (Li et al., 2025; Bergsma et al., 2025a; Shuai et al., 2024) suggest that the optimal batch size grows with dataset size and have proposed scaling laws to support this claim. However, when we compare the scaling laws of loss between $B = 512$ and $B = 8192$ (Figure 6 (d)), the curve for the large-batch setting never surpasses that of the small-batch setting, which make us rethink this viewpoint.

We suspect that the failure of the proposed scaling law arises from the limited range of fitting data. For example, Bergsma et al. (2025a) test the law using their largest batch size of fewer than 1,000 sequences ($\sim$2M tokens) and fit the scaling trend using even smaller batch sizes (Figure 1 therein). Similarly, Li et al. (2025) use a maximum batch size of fewer than 2M tokens for fitting (Figure 5(a) therein). Yet a batch size of 2M tokens (2,048 sequences in our setup) is still considered small under the GNS criterion (Figure 2). Thus, scaling rules derived from small-batch regimes may not generalize to the large-batch regime.

**The validity of the square-root LR-BS scaling rule.** The square-root LR-BS scaling rule (Malladi et al., 2022; Merrill et al., 2025) also fails in our case. On one hand, according to the scaling law of the optimal LR (Figure 6(a)), the optimal LR for large BS is lower than that for the small one, indicating that this rule does not apply in the large-batch regime under the GNS criterion. This observation is also consistent with the surge phenomenon of Adam optimizer (Li et al., 2024; Su, 2025b). Other empirical studies also support this point (Marek et al., 2025; Blake et al., 2025).

On the other hand, Figure 14 shows the loss curves for small BS, as well as for batch size schedulers with and without the square-root scaling rule. The curve following this rule not only fails to preserve the loss trajectory of the small-batch case, but also underperforms overall, converging to a higher loss

than the scheduler without this rule. Therefore, we do not incorporate this scaling rule into our batch size scheduler. Instead, we find that fixing the learning rate while ramping up the batch size yields good results.

**Batch size scheduler unlocks effective large-batch training.** We find that large-batch training, when combined with a batch size scheduler, offers many advantages over regular large-batch training. First, the batch size scheduler robustly improves the performance of large batches by a wide margin, even surpassing small batches in the data-limited regime (Figure 6 (d)). We also note that the scaling laws of hyperparameters for the batch size scheduler lie between those of small and large BS, which is intuitively natural. However, the exponent for WD in particular shows a significant gap. We attribute this discrepancy to the period immediately after batch size ramping, where residual effects from small-batch training are not yet fully eliminated. Finally, the batch size scheduler increases the GNS compared to naive large-batch training (Figure 2(b),(c)), which can be regarded as an implicit advantage. This suggests that *when we must use a large $B$ for training efficiency, we should always use a batch size scheduler*.

## 5 RELATED WORK

**Hyperparameter scaling laws.** Beyond the well-studied scaling laws of loss w.r.t. model and data size (Kaplan et al., 2020), and compute-optimal scaling laws (Hoffmann et al., 2022), researchers have begun to investigate hyperparameter scaling laws: the relationships between optimal hyperparameters and independent factors such as model size, data size, compute, or even the loss itself.

For **batch size**, two main types of scaling laws have been studied. The first is batch size–loss scaling (Hu et al., 2024; Kaplan et al., 2020; Li et al., 2024; McCandlish et al., 2018; MiniMax et al., 2025; Wang et al., 2024; Shuai et al., 2024), which describes how performance changes with batch size but does not predict the optimal batch size a priori. The second line of work characterizes the optimal batch size as a function of data or compute (Bergsma et al., 2025a; DeepSeek-AI, 2024; Li et al., 2025; Porian et al., 2024; Zhang et al., 2025), typically finding that it grows with dataset size.

For **learning rate**, several studies examine how the optimal value depends on model size, data, or compute (Porian et al., 2024; Bjorck et al., 2025; DeepSeek-AI, 2024; Li et al., 2025). However, results are sometimes contradictory: for instance, while Bjorck et al. (2025) report that the optimal learning rate decreases with data size, Li et al. (2025) argue that it increases.

For **weight decay**, attention has only recently grown. Wang & Aitchison (2025) propose, from an EMA perspective, that weight decay should decrease inversely with data size. Building on this, Bergsma et al. (2025a) suggest that EMA should decay as tokens-per-parameter (TPP) increases, implying that weight decay should decrease sublinearly with data size.

## 6 CONCLUSION

In this work, we identify trajectory invariance and its shift during training. Building on this observation, we propose new principles for efficient hyperparameter tuning. We also conduct data-based scaling law analyses to refine or challenge existing viewpoints. Our hope is that this work provides a fresh perspective on scaling laws research, helping to uncover hidden structure in the higher-dimensional hyperparameter space and guiding more efficient training practices.

**Limitations.** Our study has several limitations. First, we only experiment at a limited model scale (164M). Prior work has shown scaling collapse (Qiu et al., 2025), where loss curves from different model sizes overlap; thus, we expect trajectory invariance to extend to larger models as well. However, we are not yet able to determine how the optimal LR, WD, and ELR evolve with model size.

Second, while we analyze key hyperparameters such as $B$, $\eta$, and $\lambda$, many others remain unexplored. Important optimizer parameters like $\beta_1$, $\beta_2$, and $\varepsilon$ in Adam also influence training. For example, prior work shows that higher $\beta_2$ significantly improves performance for small batch sizes (Zhang et al., 2025; Porian et al., 2024). Since we already establish that small batch sizes outperform large ones, tuning $\beta_2$ does not alter our main conclusions. Nevertheless, we expect the trajectory invariance between LR and WD to persist under different $\beta_2$ values, and we view incorporating $\beta_2$ into an invariance formulation as a promising direction for future work.

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

# Appendix

## CONTENTS

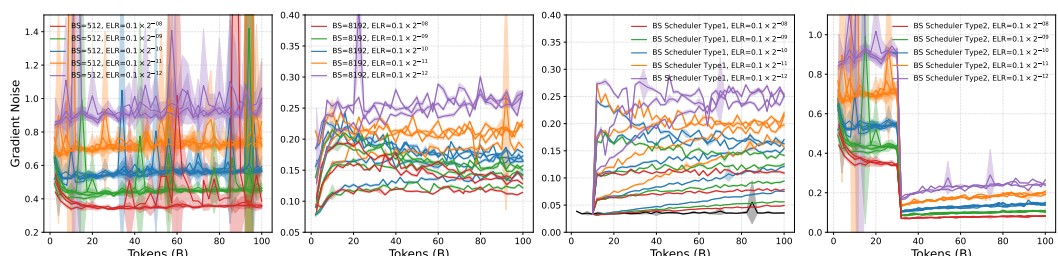

Figure 7: **Dynamics of gradient noise.**

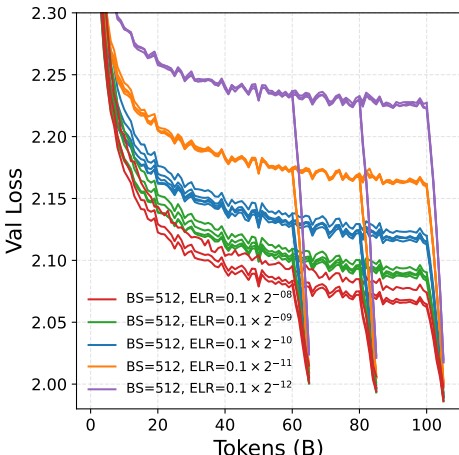

Figure 8: Trajectory invariance w.r.t ELR for WSD schedulers.

## A  MORE RESULTS

### A.1  TRAJECTORY INVARIANCE FOR GRADIENT NOISE, GRADIENT NORM, AND GRADIENT NOISE SCALE.

Here, the actual quantities we care about are $(\eta/B) \cdot \text{Tr}(\mathbf{P}^{-1}\mathbf{\Sigma})$, $\eta \cdot \|\mathbf{P}^{-1/2}\mathbf{g}\|^2$ and $\mathcal{B}_{\text{precond}}$. For clarity, we refer to these as gradient noise, gradient norm, and GNS, respectively.

In Figure 7, we show the gradient noise dynamics under various ELRs. For all runs, gradient noise becomes stable in the early stage and slowly increases as training proceeds, which indicates the gain from LR decay is increasing. Also, larger ELR gives a larger gain in LR decay. About the collapse phenomena, gradient noise exhibits earlier collapse compared loss curves, and the collapse quality is not affected by ELR.

In addition to gradient noise, gradient norm is also an important optimization metric. We show the dynamics of gradient norm in the full dynamics section A.4. Although gradient norm is not stable throughout training, the cluster pattern is still very clear.

Additionally, trajectory invariance for gradient noise scale is shown in Figure 28.

### A.2  TRAJECTORY INVARIANCE FOR WSD SCHEDULERS

See Fig. 8. We decay learning rate linearly in 5B tokens starting from 60B, 80B, and 100B tokens. Results show that trajectory invariance w.r.t. ELR continues to hold with learning rate decay.

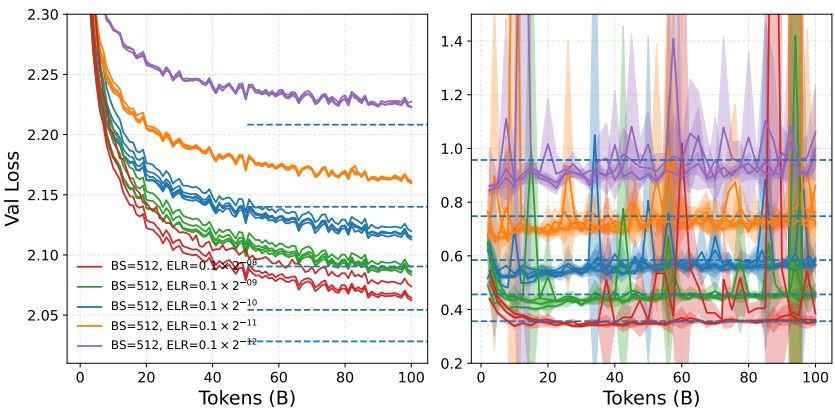

Figure 9: Fitting results.

## A.3 FITTING RESULTS

See predicted values in Figure 9. The dotted lines represent the predicted final loss $L_{0,\eta,\lambda}$, i.e., training for infinite time with constant LR $\eta$ and WD $\lambda$.

## A.4 FULL DYNAMICS

See full dynamics for $B = 512$, $B = 8192$, Type 1 BS scheduler, and Type 2 BS scheduler in Figure 10, 11, 12, 13, respectively.

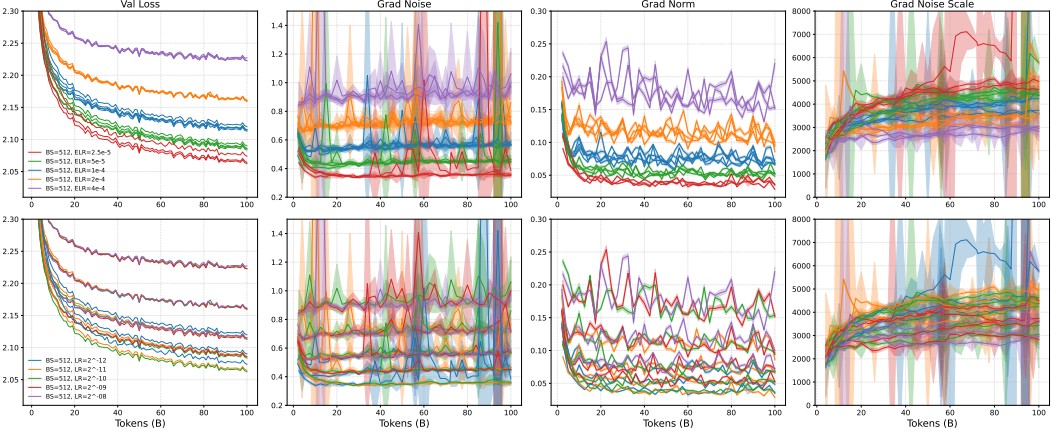

Figure 10: **Full dynamics with $B = 512$.**

## A.5 QUANTITATIVE RESULTS

We show quantitative results for loss in Figure 16 (a), Figure 17 (a); for gradient noise in Figure 18 (a), Figure 20 (a); and for gradient norm in Figure 19 (a), Figure 21 (a). All results show the collapse along ELR at small BSs and are consistent with the training dynamics presented before.

## A.6 SCALING LAWS

See Figure 23 for scaling laws under all setups, including $B = 512$, $B = 8192$, batch size scheduler Type 1, and batch size scheduler Type 2, which is more detailed version of Figure 6.

See Figure 22 for non-fitted optimal hyperparameters, i.e., optimal hyperparameters from grid search, irrelevant to the fitted quadratic functions.

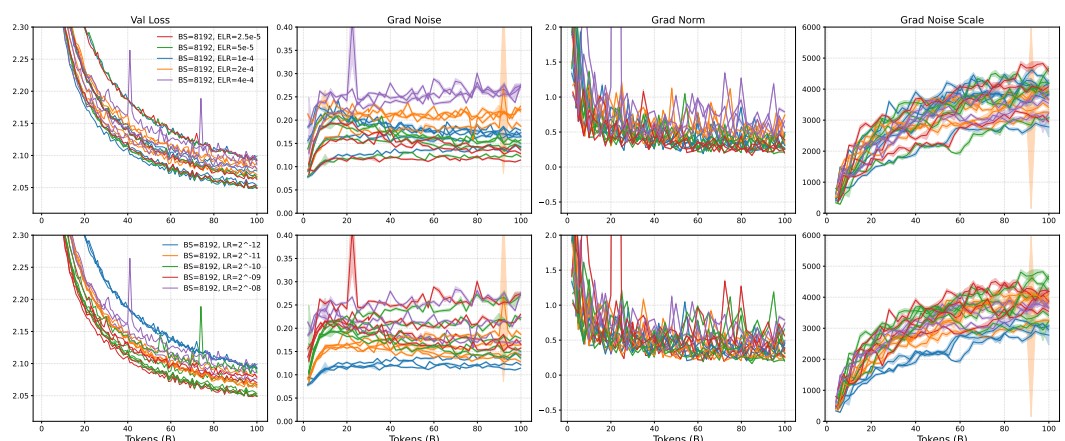

Figure 11: **Full dynamics with** $B = 8192$.

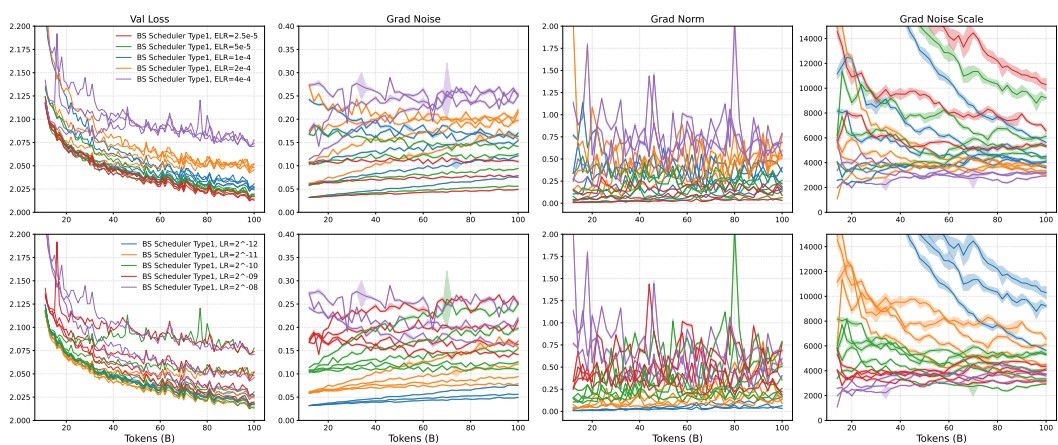

Figure 12: **Full dynamics with BS scheduler Type 1.**

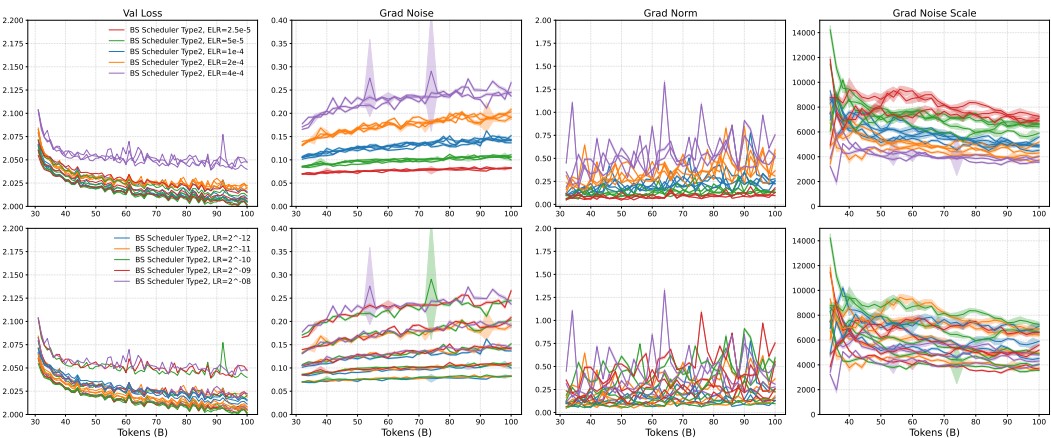

Figure 13: **Full dynamics with BS scheduler Type 2.**

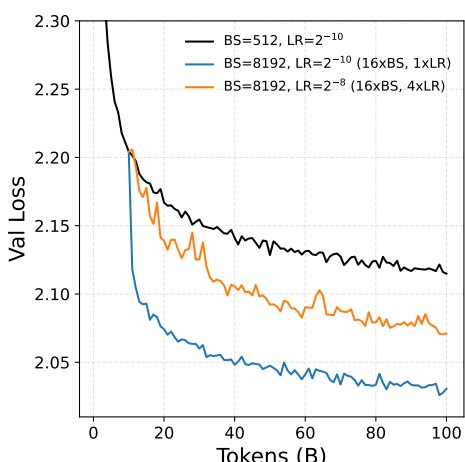

Figure 14: Square-root LR-BS scaling does not work

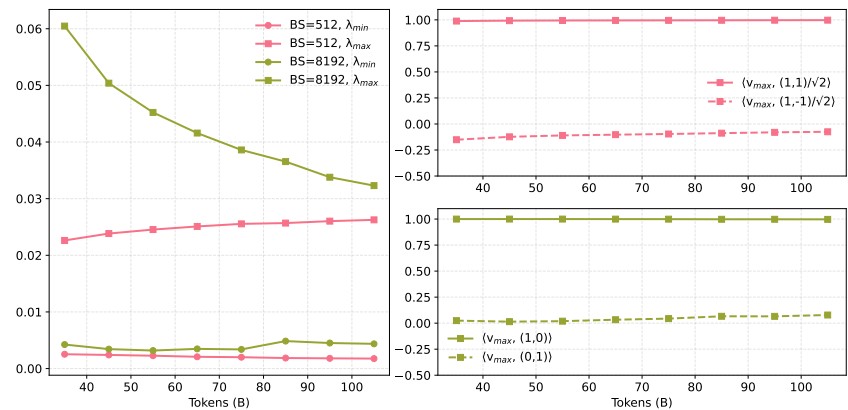

Figure 15: Eigen dynamics

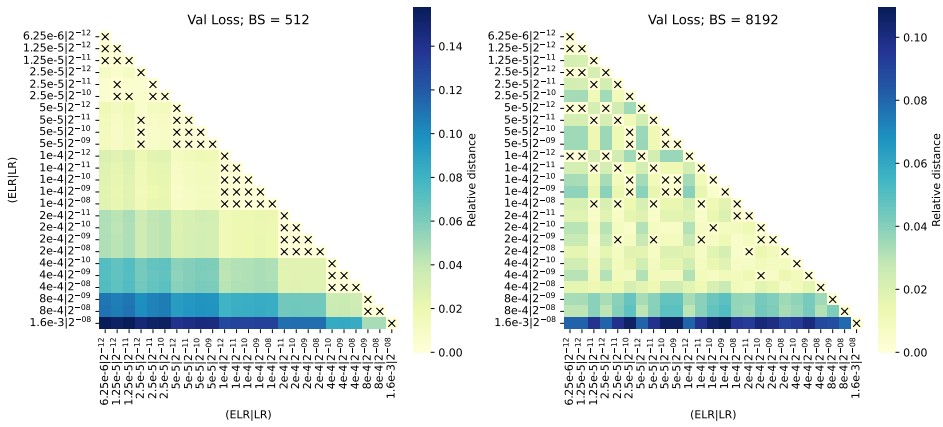

Figure 16: **Pairwise relative distance for Val Loss (by ELR).**

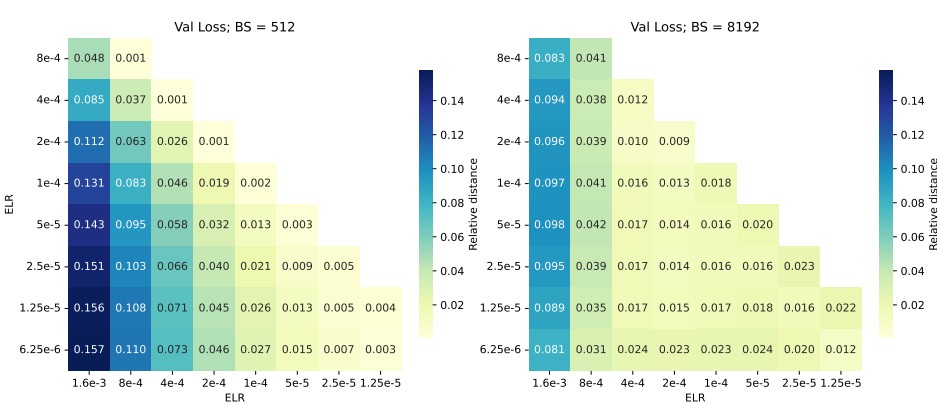

Figure 17: **Average relative distance for Val Loss (by ELR).**

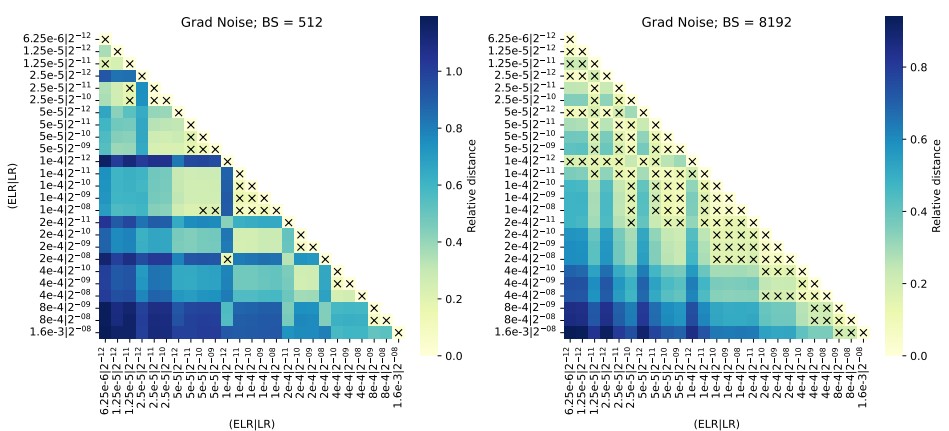

Figure 18: Pairwise relative distance for gradient noise (by ELR).

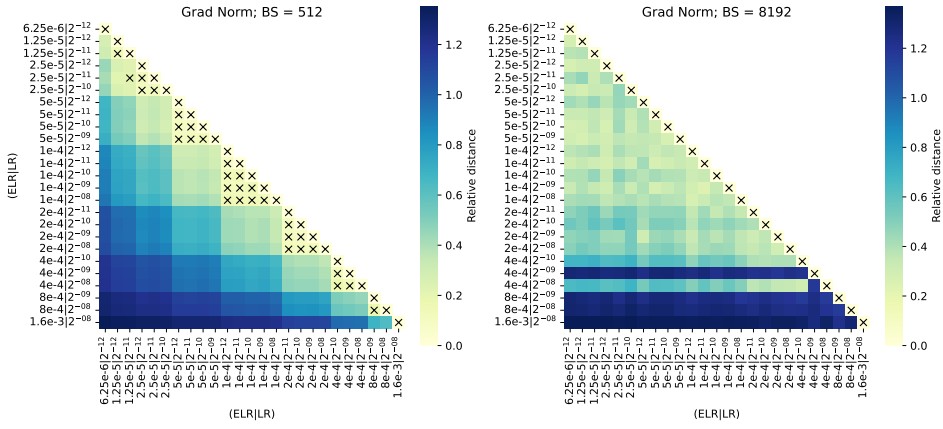

Figure 19: Pairwise relative distance for gradient norm (by ELR).

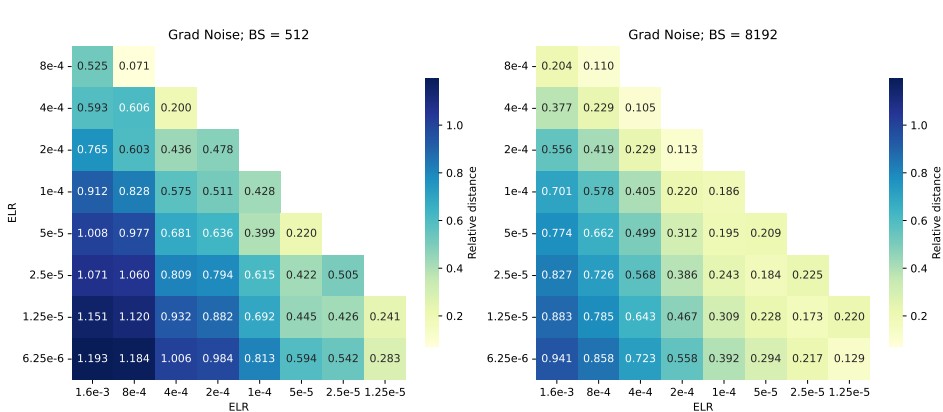

Figure 20: Average relative distance for gradient noise (by ELR).

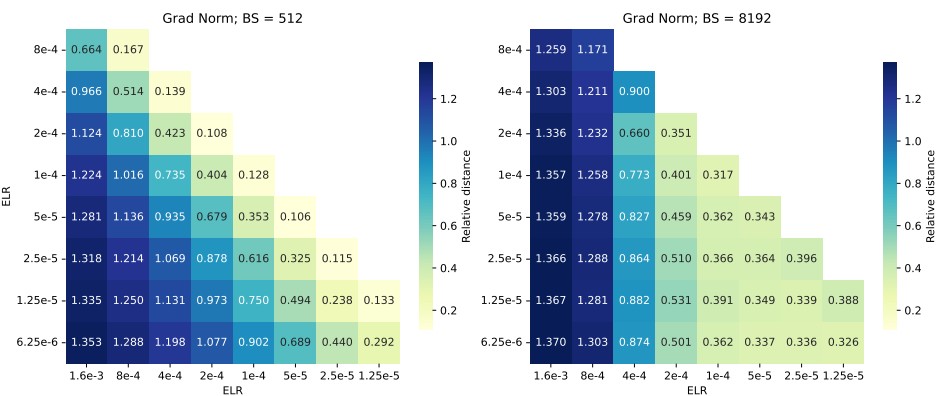

Figure 21: Average relative distance for gradient norm (by ELR).

See Figure 24 for scaling laws at fixed weight decay $\lambda = 0.1$. Implementation details are in Appendix. B.2.

**Differences in the setup for fitting scaling laws.** In Table 2 and Table 3, we compare the main differences in the setup for fitting LR scaling laws between our work and previous studies (Bjorck et al., 2025; Li et al., 2025; DeepSeek-AI, 2024). Recall that, for LR scaling laws, Bjorck et al. (2025), DeepSeek-AI (2024), and our experiments with fixed weight decay (Figure 24) yield a decreasing scaling trend; Li et al. (2025) report an increasing trend; and our experiments with tuned weight decay yield an approximately constant trend (Figure 6). We highlight three primary differences that may lead to different conclusions:

- **Tuned weight decay.** Compared with previous work, we fit LR scaling laws with a *tuned* weight decay, whereas previous works consistently use a fixed weight decay of 0.1. This is the key reason we obtain different conclusions.

- **Final learning rate.** Bjorck et al. (2025); DeepSeek-AI (2024) use a fixed ratio of max LR as the final LR, i.e., $0.1\times$ max LR, while Li et al. (2025) use a fixed final LR of 1e-5. When the max LR changes, the ratio between the final LR and the max LR therefore changes. In contrast, we use a final LR of 0, which can also be seen as a fixed ratio of max LR with ratio equal to 0. We suspect that the difference in conclusions between decreasing scaling (Bjorck et al., 2025; DeepSeek-AI, 2024) and increasing scaling (Li et al., 2025) arises from this difference.

- **Tuned batch size.** Our work and Bjorck et al. (2025) fit LR scaling laws independently for each batch size, while Li et al. (2025); DeepSeek-AI (2024) use a tuned batch size. However, the tuned batch size will probably not affect the conclusions. For example, DeepSeek-AI (2024) obtain a similar conclusion to our Figure 24 and Bjorck et al. (2025) even when tuning the batch size.

**Reproducing the decreasing scaling trend for LR at fixed weight decay.** Recall that compared with Bjorck et al. (2025), we both fit LR scaling laws independently for each model size and batch size. However, we also *tune* the weight decay, whereas Bjorck et al. (2025) uses a fixed weight decay of 0.1.

In Figure 24, when using a fixed weight decay of 0.1, we can reproduce the decreasing scaling results in Bjorck et al. (2025). In this case, our fitted exponent of the LR scaling law is $\approx -0.45$ with $R^2 = 0.9955$. We see that:

- Our exponent for LR ($\approx -0.45$) is of the same order as in Bjorck et al. (2025) (from -0.32 to -0.70, as in Table 5 therein). This means we successfully reproduce the decreasing scaling trend.

- Our exponent for ELR ($\approx -0.45$) is of the same order as the exponent for ELR when tuning weight decay ($\approx -0.34$). This again justifies our trajectory invariance principle w.r.t. ELR, since sweeping only over LR can effectively induce a sweep over ELR.

Table 2: **Setup difference in fitting learning rate scaling laws from previous studies.** [†]LR scaling laws are fitted independently for each batch size when batch size is not tuned. [‡]Bjorck et al. (2025) fit LR scaling laws at batch size of 1M up to about 50B token budget.

| Hyperparameter | Our Figure 6 | Our Figure 24 | Bjorck et al. (2025) | Li et al. (2025) |
|---|---|---|---|---|
| learning rate | tuned | tuned | tuned | tuned |
| weight decay | tuned | 0.1 | 0.1 | 0.1 |
| batch size[†] | 0.5M, 8M | 0.5M, 8M | 0.5M, 1M[‡] | tuned |
| LR schedule | WSD | WSD | cosine | cosine |
| min LR | 0 | 0 | $0.1\times$ max LR | 1e-5 |
| warmup | 1B tokens | 1B tokens | $\sim$1000 steps | 2000 steps |

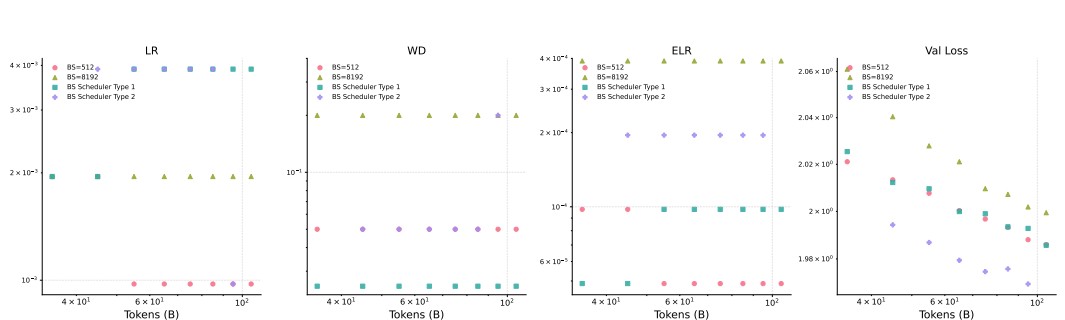

Figure 22: Non-fitted minima at each token budget

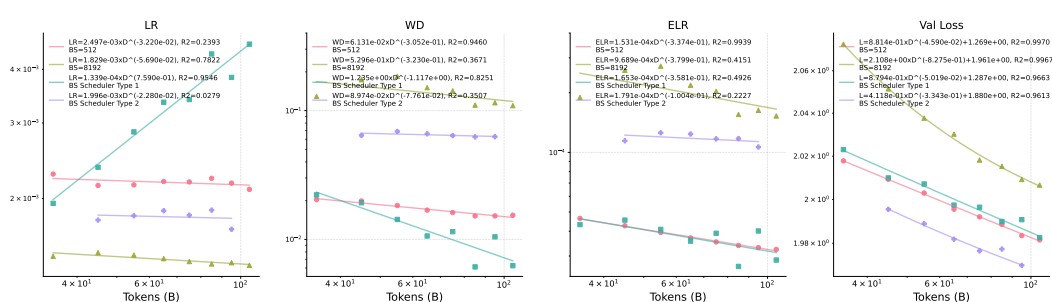

Figure 23: Full scaling laws

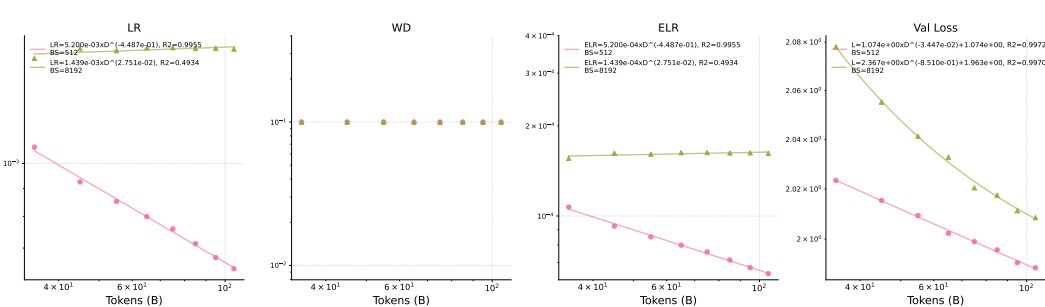

Figure 24: Scaling laws with fixed weight decay $\lambda = 0.1$.

Table 3: **Setup difference in fitting learning rate scaling laws from previous studies (Continue).**

| Hyperparameter | Our Figure 6 | Our Figure 24 | DeepSeek-AI (2024) |
|---|---|---|---|
| learning rate | tuned | tuned | tuned |
| weight decay | tuned | 0.1 | 0.1 |
| batch size | 0.5M, 8M | 0.5M, 8M | tuned |
| LR schedule | WSD | WSD | multi-step |
| min LR | 0 | 0 | $0.1\times$ max LR |
| warmup | 1B tokens | 1B tokens | 2000 steps |

## A.7 THE CONNECTION BETWEEN GRADIENT NOISE AND LEARNING RATE DECAY

Previous work shows that gradient noise is strongly correlated with the loss improvement obtained from LR decay (Qiu et al., 2025). Here, we first elaborate on this connection, then verify it in our setting, and demonstrate the agreement between LR decay and gradient noise (Figure 27).

Let $\tau := \int \eta(t)\,dt$ denote the gradient flow time, and let $\eta(\tau) \equiv \eta$ be a constant learning rate scheduler. Define $B(\tau)$, $P^{-1}\Sigma(\tau)$, and $L(\tau)$ as the batch size, preconditioned gradient noise, and loss along this constant LR schedule. Let $\eta'(\tau)$ denote the WSD scheduler with peak LR $\eta$, starting to decay at $\tau_0$, and let $B'(\tau)$, $P'^{-1}\Sigma'(\tau)$, and $L'(\tau)$ be the corresponding batch size, preconditioned gradient noise, and loss under the WSD schedule. Then, Qiu et al. (2025) show that the improvement from LR decay can be estimated as:

$$L'(\tau) - L(\tau) = \frac{1}{4}(\eta'(\tau)/B'(\tau))\operatorname{Tr}(P'^{-1}\Sigma')(\tau) - \frac{1}{4}(\eta(\tau)/B(\tau))\operatorname{Tr}(P^{-1}\Sigma)(\tau).$$

If we decay LR to zero at time $\tau$, we have

$$L'(\tau) - L(\tau) = -\frac{1}{4}(\eta(\tau)/B(\tau))\operatorname{Tr}(P^{-1}\Sigma)(\tau),$$

which means that the loss after decaying the LR to zero can be predicted from the constant LR schedule as

$$L'(\tau) = L(\tau) - \frac{1}{4}(\eta(\tau)/B(\tau))\operatorname{Tr}(P^{-1}\Sigma)(\tau). \tag{1}$$

Since both sides of Eq. (1) are computable, we evaluate the relative error in Figure 27 to verify this claim:

$$\text{relative error} := \frac{|L'(\tau) - L(\tau) + \frac{1}{4}(\eta(\tau)/B(\tau))\operatorname{Tr}(P^{-1}\Sigma)(\tau)|}{L'(\tau)},$$

## A.8 TRAJECTORY INVARIANCE IN MORE SETTINGS

**More optimizers.** We present trajectory invariance w.r.t. ELR for other optimizers, including Muon (kimi), also known as Moonlight (Liu et al., 2025), Muon (spectral) (Jordan et al., 2024), and Lion (Chen et al., 2023) (one type of SignSGD). The difference between Muon (kimi) and Muon (spectral) is solely in the adjusted LR ratio, where Muon (kimi) uses $0.2\sqrt{\max(d_{\text{in}}, d_{\text{out}})}$, and Muon (spectral) uses $\sqrt{\max(1, d_{\text{out}}/d_{\text{in}})}$.

Results are shown in Figure 25. We see that all optimizers in our experiments successfully exhibit trajectory invariance w.r.t. ELR in terms of loss curves.

**Larger models.** We show trajectory invariance w.r.t. ELR for 1B models trained up to 45B tokens in Figure 26 (a). Due to limited compute, we pick 3 combinations of (LR, WD) and keep ELR fixed. We see that 1B models successfully exhibit trajectory invariance w.r.t. ELR in terms of loss curves. The time at which the curves overlap does not lag behind that of the 164M models.

**Different $\beta_2$ in AdamW.** We show trajectory invariance w.r.t. ELR for a different value of $\beta_2 = 0.995$ in AdamW in Figure 26 (b). We see that AdamW with a different $\beta_2$ successfully exhibits trajectory invariance w.r.t. ELR in terms of loss curves.

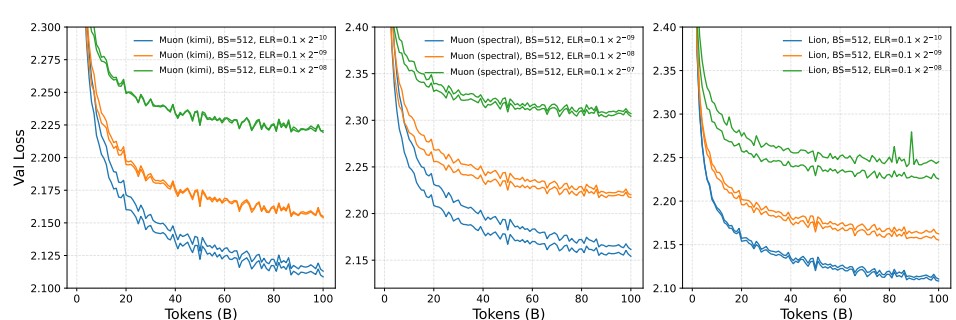

Figure 25: Trajectory invariance w.r.t. ELR for Muon and Lion.

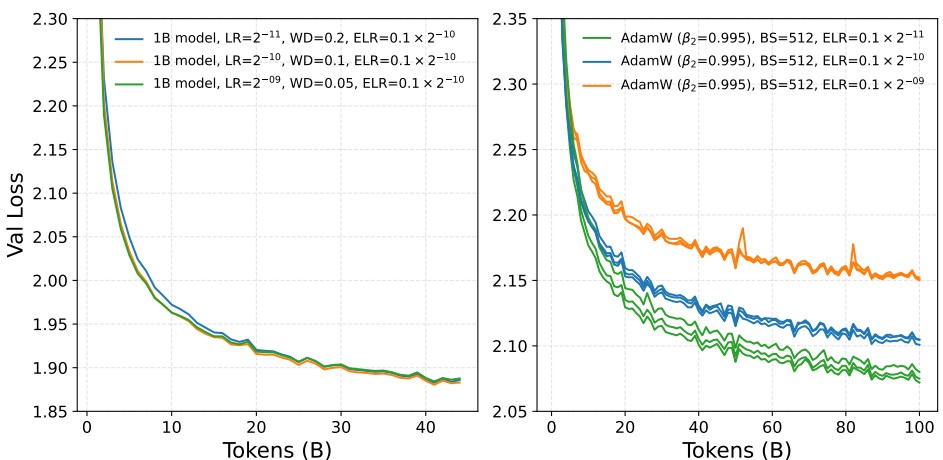

Figure 26: Trajectory invariance w.r.t. ELR for 1B model and $\beta_2 = 0.995$ in AdamW.

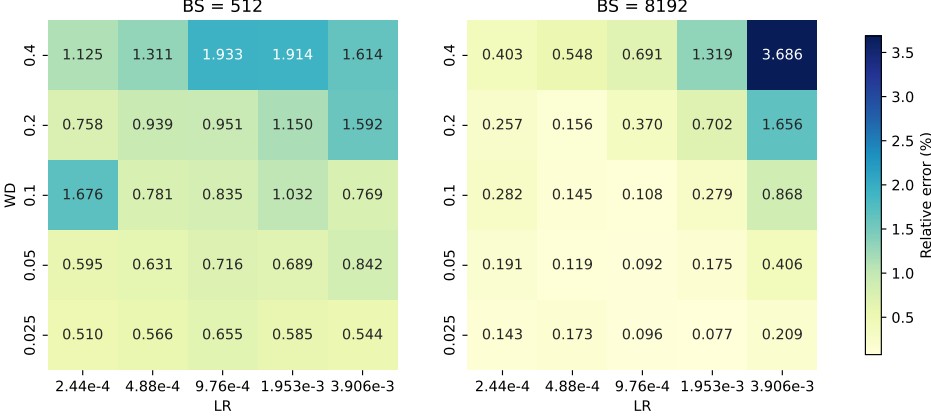

Figure 27: **Compare LR decay and gradient noise.**

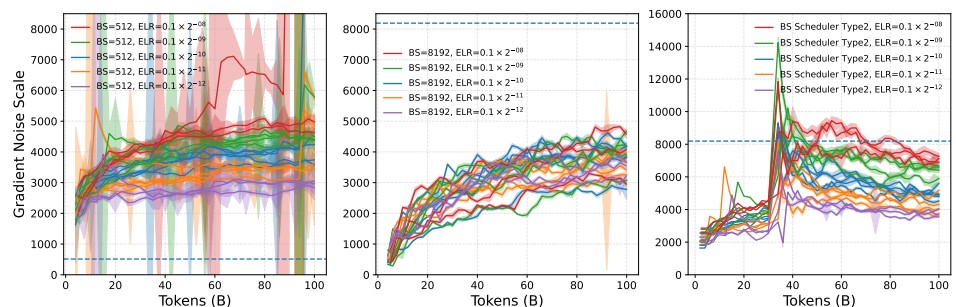

Figure 28: **Dynamics of gradient noise scale.** Legends follow Fig. 1(b).

# B IMPLEMENTATION DETAILS

## B.1 OPTIMIZATION-RELATED STATISTICS

In this section, we describe how we compute the (preconditioned) gradient noise covariance and the population gradient norm.

Recall the definitions:

$$\mathbf{g} := \mathbb{E}_{\mathbf{x}}\big[\nabla_{\mathbf{w}} L(\mathbf{w}, \mathbf{x})\big],$$
$$\mathbf{\Sigma} := \mathbb{E}_{\mathbf{x}}\big[(\tilde{\mathbf{g}}_{\mathbf{x}} - \mathbf{g})(\tilde{\mathbf{g}}_{\mathbf{x}} - \mathbf{g})^{\top}\big],$$
$$\mathbf{P} := \text{diag}(\mathbf{v}^{1/2}).$$

We first explain how to compute **one** estimator of these quantities from a single checkpoint, which consists of the model parameters and the optimizer states.

Given the checkpoint, we sample $B$ sequences and perform forward and backward passes to obtain a mini-batch gradient $\mathbf{g}_j$. We repeat this procedure $n_{\text{fwd}}$ times to obtain

$$\mathbf{g}_1, \ldots, \mathbf{g}_{n_{\text{fwd}}}.$$

From these, we estimate the population gradient by

$$\mathbf{g}_{\text{mean}} = \frac{1}{n_{\text{fwd}}} \sum_{j=1}^{n_{\text{fwd}}} \mathbf{g}_j,$$

and the trace of the batch-size-independent covariance by

$$\text{Tr}(\mathbf{\Sigma}) = \frac{B}{n_{\text{fwd}} - 1} \sum_{j=1}^{n_{\text{fwd}}} (\mathbf{g}_j - \mathbf{g}_{\text{mean}})^{\top}(\mathbf{g}_j - \mathbf{g}_{\text{mean}}).$$

For the preconditioned quantities, we use the saved second moments in the checkpoint to define the preconditioner $\mathbf{P}$. Specifically, $\mathbf{P}$ is diagonal with $\mathbf{P}_{kk} = \sqrt{\mathbf{v}_k/(1 - \beta_2^t)} + \varepsilon$, where $t$ is number of iteration and $\beta_2$ and $\varepsilon$ are the hyparparameters of Adam used in training. Then, we form

$$\mathbf{P}^{-1/2}\mathbf{g}_j \quad \text{and} \quad \mathbf{P}^{-1/2}\mathbf{g}_{\text{mean}}.$$

Here, applying $\mathbf{P}^{-1/2}$ corresponds to element-wise multiplication of two vectors, since $\mathbf{P}$ is diagonal. We then compute the trace of the batch-size-independent preconditioned covariance as

$$\text{Tr}(\mathbf{P}^{-1}\mathbf{\Sigma}) = \frac{B}{n_{\text{fwd}} - 1} \sum_{j=1}^{n_{\text{fwd}}} \big(\mathbf{P}^{-1/2}\mathbf{g}_j - \mathbf{P}^{-1/2}\mathbf{g}_{\text{mean}}\big)^{\top}\big(\mathbf{P}^{-1/2}\mathbf{g}_j - \mathbf{P}^{-1/2}\mathbf{g}_{\text{mean}}\big).$$

To inspect and reduce estimator variance, we repeat the above procedure $n_{\text{est}}$ times, obtaining **multiple** estimators for each quantity. For example, let $\text{Tr}(\mathbf{P}^{-1}\mathbf{\Sigma})_i$ denote the $i$-th estimator of $\text{Tr}(\mathbf{P}^{-1}\mathbf{\Sigma})$ computed as above. Collecting $\{\text{Tr}(\mathbf{P}^{-1}\mathbf{\Sigma})_i\}_{i=1}^{n_{\text{est}}}$, we report their sample mean and variance in the corresponding figures.

In practice, we set $n_{\text{fwd}} = 96$, $n_{\text{est}} = 10$, $B = 16$. We make the computation more efficient by utilizing multiple GPUs and intra-node communication.

## B.2 SCALING LAWS

In this section, we describe our procedures fitting scaling laws. We firstly discuss the details in Figure 6.

We estimate the optimal LR and WD for each token budget $D$ and batch size $B$. Specifically, our goal is:

$$\eta_{D,B}, \lambda_{D,B} = \arg\min_{\eta,\lambda} L(\eta, \lambda, D, B).$$

We sweep $\eta$ and $\lambda$ over a $5 \times 5$ two-dim grid as in the main experiments. Then, we fit a quadratic function, i.e., paraboloid $L_{\text{quad},D,B}(\eta, \lambda)$, based on the 25 points from experiment using linear regression in `np.linalg.lstsq` on the first and second order features We use the optimal point of this paraboloid as the estimate of $\eta_{D,B}, \lambda_{D,B}$. Specifically, we use

$$\hat{\eta}_{D,B}, \hat{\lambda}_{D,B} = \arg\min_{\eta,\lambda} L_{\text{quad},D,B}(\eta, \lambda), \quad \hat{\gamma}_{D,B} = \hat{\eta}_{D,B}\hat{\lambda}_{D,B}.$$

Next, for each $B$, we fit the scaling laws across data size. For laws without constant terms, including $\eta, \lambda, \gamma$, we use `np.polyfit` to do linear regression in the log axis. For scaling laws with constant terms, including loss $L$, we use LBFGS algorithm in `scipy.optimize.minimize`.

In Figure 24, we further fit scaling laws at fixed weight decay $\lambda = 0.1$. The only difference from Figure 6 is in the use of $L_{\text{quad},D,B}$. In this case, we only use 5 points to fit $L_{\text{quad},D,B}$. The 5 points use varied learning rate and a fixed weight decay of 0.1.

## C INTUITIVE EXPLANATION

In this section, we provide an intuitive explanation of the trajectory invariance phenomenon, in the case of a constant learning rate, for optimizers using decoupled weight decay. The update of optimizers with decoupled weight decay can be unified as:

$$w_{t+1} = (1 - \eta\lambda)w_t + \eta\lambda\left(\frac{u_t}{\lambda}\right),$$

where $u_t$ is the update given by different optimizers. Let $g_t$ be the stochastic gradient in the current iteration. For different optimizers, $u_t$ is defined as:

- AdamW: $u_t = -\dfrac{m_t}{\sqrt{v_t} + \epsilon}$

- SGD: $u_t = -g_t$

- Muon: $u_t = -\text{NewtonSchulz}(m_t)$

- Lion: $u_t = -\text{Sign}\big(\beta_1 m_{t-1} + (1 - \beta_1)g_t\big)$

Then, the whole process by which trajectory invariance emerges is:

- The $\ell^2$ update norm $\|u_t\|$ stabilizes at the early stage of training, instead of diminishing to zero.

- The $\ell^2$ parameter norm $\|w_t\|$ increases and then becomes stable, and this stable value is proportional to $\sqrt{\eta/\lambda}$.

- When the $\ell^2$ parameter norm becomes stable, the effective learning rate (relative update size) is $\gamma = \eta/\|w\|$.

- Substituting the stable parameter norm $\|w\| \propto \sqrt{\eta/\lambda}$ gives the effective learning rate $\gamma \propto \sqrt{\lambda\eta}$. We hypothesize that the loss curves will behave similarly as long as the relative update size is invariant.

**The stabilization of $\ell^2$ update norm $\|u\|$.** Earlier this year, Liu et al. (2025) observed that AdamW's update RMS norm is usually around 0.2 to 0.4. In our experiments, we also observe that AdamW's

update RMS norm stabilizes quickly (around 10,000 steps) to around 0.2. The stabilization of the update norm also happens for Muon and the Lion optimizer. A derivation for AdamW can be found in (Su, 2025a). We will include it in the revised paper for completeness.

**The stabilization of $\ell^2$ parameter norm $\|w\|$.** The EMA formulation is a good way to understand the behavior of optimizers using decoupled weight decay. It tells us that we only average over roughly the last $1/(\eta\lambda)$ minibatch updates to the current weight iterate (Wang & Aitchison, 2025). Therefore, after the update norm stabilizes, the parameter norm can also become stable under the effect of EMA. Additionally, the speed of stabilization is controlled by the learning rate: a smaller learning rate leads to slower stabilization.

The stable value is proportional to $\sqrt{\eta/\lambda}$. Some derivations of this stable value can be found in Su (2025c); Kosson et al. (2024). We will include it in the revised paper for completeness.

**Deriving effective learning rate with stable parameter norm.** When the $\ell^2$ parameter norm becomes stable, we can view the update direction as living in the tangent space, i.e., orthogonal to the parameter direction, and rotating the parameters on a hypersphere. Therefore, the relative update, or effective learning rate, is naturally $\eta/\|w_t\|$. More specifically, we have

$$w_{t+1} = (1 - \eta\lambda)\|w_t\| \left( \frac{w_t}{\|w_t\|} + \frac{\eta}{(1 - \eta\lambda)\|w_t\|} u_t \right).$$

Using the approximation that $\|w_t\| \approx \|w_{t+1}\|$ and $1 - \eta\lambda \approx 1$, we get

$$\frac{w_{t+1}}{\|w_{t+1}\|} \approx \frac{w_t}{\|w_t\|} + \frac{\eta}{\|w_t\|} u_t.$$

In summary, when the parameter norm is stable, the effective learning rate is $\eta/\|w\|$.

## LLM USAGE

We used advanced language models solely as writing assistants during manuscript preparation. Their role was limited to grammar correction, clarity improvement, and refining sentence flow, without altering the intended meaning. All research ideas, methods, and results are entirely the work of the authors.

