# OpenReview forum: "Efficient Hyperparameter Tuning via Trajectory Invariance Principle"
_ICLR.cc/2026/Conference — Submitted to ICLR 2026_

### Official Review · Reviewer_qowG · 2025-10-30

**Soundness:** 1
**Presentation:** 2
**Contribution:** 1
**Rating:** 2
**Confidence:** 4

**Summary:**

The paper introduces a trajectory invariance phenomenon in language model pretraining, where loss, gradient noise, and gradient norm curves remain nearly identical across runs sharing a composite of learning rate and weight decay. This observation suggests that the two-dimensional hyperparameter space of learning rate and weight decay can be reduced to a single tuning direction. The authors also discuss implications for hyperparameter scaling laws and efficient hyperparameter tuning strategies.

**Strengths:**

- The paper identifies a trajectory invariance phenomenon in pre-training dynamics, showing that certain combinations of learning rate and weight decay yield similar training trajectories. This observation may help reduce the effective hyperparameter search space and improve tuning efficiency.

**Weaknesses:**

- The study is conducted on a single model size (164M-parameter model), meaning there is no cross-scale analysis. Without experiments on multiple model sizes (e.g., ≥1B parameters), the generality and robustness of the proposed principle cannot be established.

- The paper claims to refine or challenge existing scaling laws, but prior scaling-law research derives conclusions from cross-scale comparisons across multiple model sizes. Since this work relies on a single-scale setting, the evidence is not sufficient to support the broader implications suggested by the paper.

**Questions:**

Please refer to the Weaknesses.

---

> ### Author Response · Authors · 2025-11-21
> **Rebuttal**
>
> Thank you for your review.
>
> > W1. The study is conducted on a single model size (164M-parameter model), meaning there is no cross-scale analysis. Without experiments on multiple model sizes (e.g., ≥1B parameters), the generality and robustness of the proposed principle cannot be established.
>
> We show trajectory invariance w.r.t. ELR for 1B models trained up to 45B tokens in **Figure 26 (a) in Appendix A.8**. Due to limited compute, we pick 3 combinations of (LR, WD) and keep ELR fixed.
> We see that 1B models successfully exhibit trajectory invariance w.r.t. ELR in terms of loss curves. The time at which the curves overlap does not lag behind that of the 164M models.
>
> > W2. The paper claims to refine or challenge existing scaling laws, but prior scaling-law research derives conclusions from cross-scale comparisons across multiple model sizes. Since this work relies on a single-scale setting, the evidence is not sufficient to support the broader implications suggested by the paper.
>
> On one hand, based on experiments with 1B models, trajectory invariance continues to hold at the 1B scale. Since trajectory invariance is about a geometric property of training transformers, we expect it to hold at larger scales as well.
>
> On the other hand, our conclusions in Section 4 focus on scaling laws across dataset size. We note that in [1,2], scaling laws across dataset size are consistent across different model sizes (Figure 4 in [1] and Figure 5 in [2]). Therefore, we expect that our conclusions at least hold at the 1B scale.
>
> ---
> [1] Scaling Optimal LR Across Token Horizons.
>
> [2] Predictable scale: Part i–optimal hyperparameter
> scaling law in large language model pretraining.

---

> ### Author Response · Authors · 2025-11-28
> **Looking forward your feedback**
>
> Dear Reviewer qowG,
>
> We hope that our responses adequately address your concerns. As the deadline of this discussion phase is approaching, we warmly welcome further discussion regarding any additional concerns that you may have.
>
> Thank you for the time and appreciation that you have dedicated to our work.
>
> Best regards,
>
> Authors of submission 5475

---

> > ### Comment · Reviewer_qowG · 2025-11-28
> >
> > We thank the authors for the rebuttal and the additional experiments. The added theoretical intuition and the expanded experiments involving a larger model size and multiple optimizers are appreciated.
> >
> >
> > However, despite these improvements, the core concerns remain largely unaddressed. The experimental evidence remains insufficient for the claimed scaling-law refinement, as the experiments cover only a single model scale with very limited exploration at 1B. This narrow scope makes it difficult to assess whether the observed behavior generalizes or meaningfully refines existing scaling laws.
> >
> >
> > Therefore, I will maintain my original score.

---

> ### Author Response · Authors · 2025-11-28
>
> We thank reviewer qowG again for the prompt follow-up and for carefully considering our rebuttal and additional experiments.
>
> Our primary goal in this work is to identify a general training principle. For this reason, we deliberately chose to allocate our computational budget to a dense and systematic exploration at smaller scales, instead of spreading the same budget thinly over many larger models.
>
> Concretely, the total compute used in our experiments is at least 1.6e21 FLOPs, which corresponds to roughly 19.2K GPU hours. This is a substantial budget, and concentrating it on smaller models allows us to characterize the phenomenon—especially **trajectory invariance to effective learning rate (ELR)**—with much higher fidelity and confidence than if we had run a few sparse configurations at larger scales.
>
> We should allocate compute more intelligently and efficiently, instead of pursuing scaling for its own sake, as Ilya emphasized in his recent talk.
>
> Given that **trajectory invariance to ELR** is a property related to training geometry, we have provided a strong intuitive explanation, and based on our own 1B results, we have reason to believe it is generally applicable at larger scales.
>
> We also want to mention that several prior works on scaling laws conduct their main analyses at moderate model scales when the focus is not on model size itself. For example:
> – [1] studies how to use scaling laws to choose data mixtures and fits these laws using relatively small models across multiple settings (up to 610M).
> – [2] investigates learning-rate scaling across dataset sizes and finds that the resulting scaling laws are consistent across different model sizes (50M vs 760M).
>
> Overall, we believe our experimental design and explanation provide strong evidence for a general and scale-robust training principle, even if a full sweep at very large scales is left for future work.
>
> ---
> [1] Scaling Laws for Optimal Data Mixtures
>
> [2] Scaling Optimal LR Across Token Horizons

---

### Official Review · Reviewer_vF4o · 2025-10-31

**Soundness:** 3
**Presentation:** 4
**Contribution:** 3
**Rating:** 8
**Confidence:** 3

**Summary:**

The paper presents how the space of hyperparameters in llm optimization may be simplified by considering solely the effective learning rate (product of learning rate and weight decay) rather than learning rate and weight decay separated. This claim is corroborated through numerous experiments that help understand its relevance and its limitations in different situations. Namely, the authors show that for a small number of iterations (independent of batch size interestingly), losses overlap for a given learning rate and not the effective learning rate. This fact guides the authors to try batch size schedulers that help retrieve invariance of the losses with effective learning rates. In general the experimental results, looed through many faces help revise and question our understanding of some learning rate scaling rules such as the square root LR-BS scaling rule.

**Strengths:**

- The paper is very well presented and written. Key concepts are introduced early. Each experiment is well thought and illustrates well the claims.
- The relevance of the effective leraning rate (ELR) is presented under many facets: overlapping losses, pairwise relative distances, optimal direction in the hyperparameter search space.
- The authors fit some scaling laws with effective learning rate that could be reused by the community.
- The authors also show that gradient noise can overlap with effective learning rate.
- The batch-size scheduling appears new to me and is a very interesting approach to capture benefits of various batch size scales.
- Discussion of related work is well done. The reader understands that similar observations were done and understands how this paper differs.

**Weaknesses:**

- It is unclear whether the effects are observed at different model sizes.
- The experiments are limited to a single optimizer. SignSGD or Muon for example would be great candidates to explore the claims.

**Questions:**

- The authors point that previous studies found different conclusions in terms of scaling laws for example. How is it possible? What changed between these studies?
- Can the authors test their claim on another optimizer?

---

> ### Author Response · Authors · 2025-11-21
> **Rebuttal**
>
> Thank you for your positive review and for acknowledging the strengths of our paper. Regarding your questions:
>
> > W1. It is unclear whether the effects are observed at different model sizes.
>
> We show trajectory invariance w.r.t. ELR for 1B models trained up to 45B tokens in **Figure 26 (a) in Appendix A.8**. Due to limited compute, we pick 3 combinations of (LR, WD) and keep ELR fixed.
> We see that 1B models successfully exhibit trajectory invariance w.r.t. ELR in terms of loss curves. The time at which the curves overlap does not lag behind that of the 164M models.
>
> > W2 + Q2. The experiments are limited to a single optimizer. SignSGD or Muon for example would be great candidates to explore the claims.
>
> We present trajectory invariance w.r.t. ELR for other optimizers, including Muon (kimi), also known as Moonlight, Muon (spectral), and Lion. The difference between Muon (kimi) and Muon (spectral) is solely in the adjusted LR ratio.
>
> Results are shown in **Figure 25 in Appendix A.8**. We see that all optimizers in our experiments successfully exhibit trajectory invariance w.r.t. ELR in terms of loss curves.
>
>
> > Q1. The authors point that previous studies found different conclusions in terms of scaling laws for example. How is it possible? What changed between these studies?
>
> Thank you for asking this question.
>
> We compare the main differences in the setup for fitting LR scaling laws between our work and previous studies. Please refer to the **Table 2 and Table 3 in Appendix A.6 of the revised paper**.
>
> Recall that, for LR scaling laws, [1,2], and our experiments with fixed weight decay (Figure 24) yield a decreasing scaling trend; [3] report an increasing trend; and our experiments with tuned weight decay yield an approximately constant trend (Figure 6).
> We highlight three primary differences that may lead to different conclusions, including tuned weight decay, final learning rate, and tuned batch size.
>
> In simple terms:
> - We attribute the difference between the decreasing and constant trends to whether weight decay is tuned. In fact, we can reproduce the decreasing scaling trend with a fixed weight decay (Figure 24 in Appendix A.6).
> - We suspect that the difference between the decreasing and increasing trends is due to the choice of final learning rate (fixed ratio of max LR vs. fixed final LR).
> - We suspect that tuning the batch size does not affect these conclusions.
>
> Please refer to Appendix A.6 for a detailed discussion.
>
> ---
> [1] Scaling Optimal LR Across Token Horizons.
>
> [2] Deepseek llm: Scaling open-source language models with longtermism.
>
> [3] Predictable scale: Part i–optimal hyperparameter
> scaling law in large language model pretraining.

---

> > ### Author Response · Authors · 2025-11-28
> > **Looking forward your feedback**
> >
> > Dear Reviewer vF4o,
> >
> > We hope that our responses adequately address your concerns. As the deadline of this discussion phase is approaching, we warmly welcome further discussion regarding any additional concerns that you may have.
> >
> > Thank you for the time and appreciation that you have dedicated to our work.
> >
> > Best regards,
> >
> > Authors of submission 5475

---

### Official Review · Reviewer_zBM1 · 2025-11-02

**Soundness:** 3
**Presentation:** 1
**Contribution:** 1
**Rating:** 2
**Confidence:** 3

**Summary:**

This paper introduces "trajectory invariance," a phenomenon in deep learning pre-training where loss curves and other metrics closely overlap. Early in training, this invariance is with respect to the learning rate (LR), meaning runs with the same LR but different weight decays (WDs) follow similar paths. As training progresses, particularly with a sufficient number of iterations, this invariance shifts to the effective learning rate (ELR), defined as the product of LR and WD. The authors demonstrate that this principle effectively reduces the two-dimensional tuning space of LR and WD to a single dimension, proposing a more efficient tuning strategy: tune along the "salient direction" (ELR in small-batch or sufficient-iteration regimes, and LR in large-batch regimes) to optimize performance. The work also refines existing hyperparameter scaling laws, challenging prior viewpoints on optimal batch size and learning rate scaling.

**Strengths:**

1. The paper introduces a novel scaling law for hyperparameter tuning, providing insights for practical applications.
2. The comprehensive experiments and discussions indicate that the proposed principle demonstrates some robustness.

**Weaknesses:**

1. The paper is poorly written and falls significantly below the standards expected of a top-tier conference. Many concepts are introduced before being properly defined (even without definition). The figures are also of very low quality, curves often overlap and are difficult to distinguish. The figure legends lack consistency; for instance, in Figure 1, the left subfigure uses “2^-12” (which is already too casual for scientific publications) while the right subfigure uses “2e−4.” Additionally, Figure 3’s caption redundantly includes “For example, For example”. The appendix appears incomplete, particularly Sections A.2 and A.3.
2. There is a lack of analytical discussion regarding this invariance phenomenon. For instance, insights can already be drawn from the update rule of algorithms incorporating weight decay. A typical update takes the form $w_{t+1} = (1 - \eta \lambda) w_t + \eta g_t$,
where $g_t$ denotes a gradient estimator, such as the stochastic gradient in SGD or $m_t / (\sqrt{v_t} + \epsilon)$ in Adam. From this expression, it becomes intuitive that early in training, the gradient magnitude is relatively large and thus dominates, making the learning rate $\eta$ the more influential factor. As training progresses and the gradients tend to converge, the first term with coefficient $(1 - \eta \lambda)
$ becomes dominant, thereby amplifying the relative effect of the factor $\eta \lambda$.
3. The paper makes rather strong claims, such as “we find that scaling law for LR is wrong.” The authors should provide sufficient justification for such assertions. Are the experimental findings reported in existing scaling law studies incorrect, or do those conclusions simply not apply under the specific conditions of the authors’ setup? Does this also imply that the authors’ findings lack generality and may not extend to the experimental setups used in those prior studies?
4. The generality of the proposed invariance remains uncertain. How does this phenomenon manifest across different optimizers, tasks, and network architectures? Moreover, can the same behavior be consistently observed when employing various deep learning techniques, such as different forms of normalization?
5. As an empirical study, the paper omits certain important implementation details. For instance, how is $\Sigma$ computed in practice? Is it estimated using an additional sampled batch? What batch size is used for this estimation, and how reliable or accurate is the resulting approximation?

**Questions:**

Please refer to the weaknesses section.

---

> ### Author Response · Authors · 2025-11-21
> **Rebuttal (Part 1)**
>
> Thank you for your review and for acknowledging the strengths of our paper. Regarding your questions:
>
>
> > W1. The paper is poorly written and falls significantly below the standards expected of a top-tier conference. Many concepts are introduced before being properly defined (even without definition). The figures are also of very low quality, curves often overlap and are difficult to distinguish. The figure legends lack consistency; for instance, in Figure 1, the left subfigure uses "2^-12" (which is already too casual for scientific publications) while the right subfigure uses "2e−4." Additionally, Figure 3’s caption redundantly includes “For example, For example”. The appendix appears incomplete, particularly Sections A.2 and A.3
>
> Thank you for the suggestion.
> - We revise the legends in the main figures and reduced the token horizon in Figure 1(a) to improve clarity.
> - We replace Figure 2 with a simpler verison for clearness and move the full version to the appendix.
> - We expand Appendix A.2 and A.3.
> - We correct the typo in the caption of Figure 3.
> - Regarding unclear concepts: could you please indicate which concepts you feel are used before being properly defined?
>
>
> > W3. The paper makes rather strong claims, such as “we find that scaling law for LR is wrong.” The authors should provide sufficient justification for such assertions. Are the experimental findings reported in existing scaling law studies incorrect, or do those conclusions simply not apply under the specific conditions of the authors’ setup? Does this also imply that the authors’ findings lack generality and may not extend to the experimental setups used in those prior studies?
>
> Thank you for asking this question.
>
> We would like to clarify that our findings do not imply that experimental results in existing studies are "incorrect"; rather, they indicate that those results are conditional on a **narrower** setup. In contrast, we want to stress that when we study a **broader** range of the hyperparameter space, we obtain a different and more general conclusion. This means that our results are **not** an artifact of specific conditions, such as a particular optimizer or hyperparameter choice.
> In fact, when we fit scaling laws using a limited range of hyperparameters as in previous studies, we can indeed reproduce their results. Below, we elaborate on these differences and provide supporting evidence.
>
> We first discuss the differences in the setup for fitting scaling laws, i.e., what additional factors we consider. Specifically, compared with [1], we both fit LR scaling laws independently for each model size and batch size. However, we also **tune** the weight decay, whereas [1] uses a fixed weight decay of 0.1, which is the key reason we obtain different conclusions.
> We provide a detailed comparison of setups in **Table 2 and Table 3 (Appendix A.6 of the revised paper)**.
>
> In Figure 24 (Appendix A.6 of the revised paper), when using a fixed weight decay of 0.1, we can reproduce the decreasing scaling results in [1]. In this case, our fitted exponent of the LR scaling law is $\approx -0.45$ with $R^2 = 0.9955$. We see that:
>
> - Our exponent for LR ($\approx -0.45$) is of the same order as in [1] (from -0.32 to -0.70, as in Table 5 therein). This means we successfully reproduce the decreasing scaling trend.
> - Our exponent for ELR ($\approx -0.45$) is of the same order as the exponent for ELR when tuning weight decay ($\approx -0.34$). This again justifies our trajectory invariance principle w.r.t. ELR, since sweeping only over LR can effectively induce a sweep over ELR.
>
> Finally, we acknowledge that some of our wording, such as "is wrong," was a bit too strong, and we have softened these expressions in the revised version. We also acknowledge that our scaling results may not hold in more general settings. Our main intention with that sentence is simply to convey that we obtain different scaling behavior when considering a more general setup.

---

> ### Author Response · Authors · 2025-11-21
> **Rebuttal (Part 2)**
>
> > W2. There is a lack of analytical discussion regarding this invariance phenomenon. For instance, insights can already be drawn from the update rule of algorithms incorporating weight decay. A typical update takes the form $w_{t+1} = (1-\eta\lambda w_t) + \eta g_t$, where $g_t$ denotes a gradient estimator, such as the stochastic gradient in SGD or $m_t/\sqrt{v_t}+\epsilon$ in Adam. From this expression, it becomes intuitive that early in training, the gradient magnitude is relatively large and thus dominates, making the learning rate the more influential factor. As training progresses and the gradients tend to converge, the first term with coefficient $(1-\eta\lambda)$ becomes dominant, thereby amplifying the relative effect of the factor $\eta\lambda$.
>
> Thank you for asking this question.
>
>
> Below, we provide an intuitive explanation of the trajectory invariance phenomenon, in the case of a constant learning rate, for optimizers using decoupled weight decay. The update of optimizers with decoupled weight decay can be unified as:
> $$
>     w_{t+1} = (1-\eta\lambda)w_t + \eta\lambda (u_t/\lambda),
> $$
> where $u_t$ is the update given by different optimizers.
> Let $g_t$ be the stochastic gradient in the current iteration. For different optimizers, $u_t$ is defined as:
>
> - AdamW: $u_t = -\dfrac{m_t}{\sqrt{v_t} + \epsilon}$
> - SGD: $u_t = -g_t$
> - Muon: $u_t = -\text{NewtonSchulz}(m_t)$
> - Lion: $u_t = -\text{Sign}\bigl(\beta_1 m_{t-1} + (1-\beta_1) g_t\bigr)$
>
>
> Then, the whole process by which trajectory invariance emerges is:
> - The $\ell^2$ update norm $||u_t||$ stabilizes at the early stage of training, instead of diminishing to zero.
> - The $\ell^2$ parameter norm $||w_t||$ increases and then becomes stable, and this stable value is proportional to $\sqrt{\eta/\lambda}$.
> - When the $\ell^2$ parameter norm becomes stable, the effective learning rate (relative update size) is $\gamma = \eta/||w||$.
> - Substituting the stable parameter norm $||w|| \propto \sqrt{\eta/\lambda}$ gives the effective learning rate $\gamma \propto \sqrt{\lambda\eta}$. We hypothesize that the loss curves will behave similarly as long as the relative update size is invariant.
>
>
> **The stabilization of $\ell^2$ update norm $||u||$.**
>
> Earlier this year, [3] observed that AdamW’s update RMS norm is usually around 0.2 to 0.4. In our experiments, we also observe that AdamW’s update RMS norm stabilizes quickly (around 10,000 steps) to around 0.2.
> The stabilization of the update norm also happens for Muon and the Lion optimizer.
> A derivation for AdamW can be found in [4]. We will include it in the revised paper for completeness.
>
> **The stabilization of $\ell^2$ parameter norm $||w||$.**
>
> The EMA formulation is a good way to understand the behavior of optimizers using decoupled weight decay.
> It tells us that we only average over roughly the last $1/(\eta\lambda)$ minibatch updates to the current weight iterate [2]. Therefore, after the update norm stabilizes, the parameter norm can also become stable under the effect of EMA. Additionally, the speed of stabilization is controlled by the learning rate: a smaller learning rate leads to slower stabilization.
>
> The stable value is proportional to $\sqrt{\eta/\lambda}$.
> Some derivations of this stable value can be found in [5,6]. We will include it in the revised paper for completeness.
>
> **Deriving effective learning rate with stable parameter norm.**
>
> When the $\ell^2$ parameter norm becomes stable, we can view the update direction as living in the tangent space, i.e., orthogonal to the parameter direction, and rotating the parameters on a hypersphere. Therefore, the relative update, or effective learning rate, is naturally $\eta/|w_t|$. More specifically, we have
> $$
>     w_{t+1} = (1-\eta\lambda){||w_{t}||} \left(\frac{w_t}{{||w_{t}||}} + \frac{\eta}{{(1-\eta\lambda)||w_{t}||}} u_t\right).
> $$
> Using the approximation that ${||w_{t}||} \approx {||w_{t+1}||}$ and $1-\eta\lambda \approx 1$, we get
> $$
>     \frac{w_{t+1}}{||w_{t+1}||} \approx \frac{w_t}{{||w_{t}||}} + \frac{\eta}{{||w_{t}||}} u_t.
> $$
> In summary, when the parameter norm is stable, the effective learinng rate is $\eta/||w||$.

---

> ### Author Response · Authors · 2025-11-21
> **Rebuttal (Part 3)**
>
> > W4. The generality of the proposed invariance remains uncertain. How does this phenomenon manifest across different optimizers, tasks, and network architectures? Moreover, can the same behavior be consistently observed when employing various deep learning techniques, such as different forms of normalization?
>
> **More optimizers.** We present trajectory invariance w.r.t. ELR for other optimizers, including Muon (kimi), also known as Moonlight, Muon (spectral), and Lion. The difference between Muon (kimi) and Muon (spectral) is solely in the adjusted LR ratio.
>
> Results are shown in **Figure 25 in Appendix A.8**. We see that all optimizers in our experiments successfully exhibit trajectory invariance w.r.t. ELR in terms of loss curves.
>
> **Larger models.**
> We show trajectory invariance w.r.t. ELR for 1B models trained up to 45B tokens in **Figure 26 (a) in Appendix A.8**. Due to limited compute, we pick 3 combinations of (LR, WD) and keep ELR fixed.
> We see that 1B models successfully exhibit trajectory invariance w.r.t. ELR in terms of loss curves. The time at which the curves overlap does not lag behind that of the 164M models.
>
>
> **Different $\beta_2$ in AdamW.**
> We show trajectory invariance w.r.t. ELR for a different value of $\beta_2=0.995$ in AdamW in **Figure 26 (b) in Appendix A.8**. We see that AdamW with a different $\beta_2$ successfully exhibits trajectory invariance w.r.t. ELR in terms of loss curves.
>
> **Other tasks and architecture.**
> Evidences for SGD on image classifications are shown in Figure 3 in [7].
>
>
> > W5. As an empirical study, the paper omits certain important implementation details. For instance, how is $\Sigma$ computed in practice? Is it estimated using an additional sampled batch? What batch size is used for this estimation, and how reliable or accurate is the resulting approximation?
>
> We add one section in appendix to discuss implementation details. please refer to Appdendix B.
>
> ---
> [1] Scaling Optimal LR Across Token Horizons.
>
> [2] How to set AdamW’s weight decay as you scale model and dataset size.
>
> [3] Muon is Scalable for LLM Training.
>
> [4] https://kexue.fm/archives/11267
>
> [5] https://kexue.fm/archives/11307
>
> [6] Rotational Equilibrium: How Weight Decay Balances Learning Across Neural Networks
>
> [7] Why Do We Need Weight Decay in Modern Deep Learning?

---

> ### Author Response · Authors · 2025-11-28
> **Looking forward your feedback**
>
> Dear Reviewer zBM1,
>
> We hope that our responses adequately address your concerns. As the deadline of this discussion phase is approaching, we warmly welcome further discussion regarding any additional concerns that you may have.
>
> Thank you for the time and appreciation that you have dedicated to our work.
>
> Best regards,
>
> Authors of submission 5475

---

### Official Review · Reviewer_J8na · 2025-11-04

**Soundness:** 2
**Presentation:** 3
**Contribution:** 2
**Rating:** 4
**Confidence:** 5

**Summary:**

The paper considers the problem of hyperparameter tuning. Concretely, they propose a phenomenon in neural network training, called "trajectory invariance", which claims that the loss curves admit an invariance w.r.t. $\gamma$ - a quantity that combines the learning rate and weight decay. Based on such a phenomenon, the authors propose a promising way to tune multiple hyperparameters in practice. For example, practitioners can fix the learning rate and only tune the weight decay, turning a problem of 2 hyperparameters into one effectively.

**Strengths:**

The paper is easy to follow.

**Weaknesses:**

First things first, I am not an expert in this field, so maybe my evaluation is a bit off. However, it seems to me that the paper is over-claiming (a lot):

1. The paper claims to have discovered the "trajectory invariance principle" for efficient hyperparameter tuning. However, there is one critical thing: the paper only conducts its experiments on the AdamW optimizer (and with the standard setting as well). Note that in AdamW, the weight decay $\lambda$ is a simple, direct decay applied to the weights, __separated__ from the gradient update. Therefore, I assume that the joint effect of learning rate $\eta$ and weight decay $\lambda$ parameter might be simple, leading to a simple term ELR $\eta = \lambda \cdot \eta$ that can capture the trajectory (in some sense). However, this does not mean that the analysis can be applied to other optimizers, including Vanilla Adam, and especially LION, Muon, of which the update principles are very different.

2. The above suggests that the finding might not be a "principle", but rather an "artifact" that happens in some sense only for AdamW. In such a case, I expect a (simplified) theoretical analysis on why this phenomenon happens for AdamW (or at least some strong intuitive explanation with concrete evidence), so that the paper might be interesting in some sense. Unfortunately, this is not the case for this paper.

3. Moreover, if we reluctantly accept the experiments restricted to AdamW, I still think that the experiments are insufficient. For example, the authors explicitly acknowledge that optimizer states of AdamW, like $\beta_2$, are "important" and "influence" the training. However, the authors only consider the experiments with one value of $\beta_2$

In summary, I believe that the empirical findings in this paper are insufficient for a high-quality paper. I strongly recommend that the authors should at least validate their findings with: (1) more optimizer settings, (2) more configuration in each optimizer, ... or try to develop some (simplified) theoretical justifications for their findings, so that they can boost this paper's contribution. As of this state, I unfortunately cannot recommend an acceptance for this paper.

**Questions:**

See weaknesses

---

> ### Author Response · Authors · 2025-11-21
> **Rebuttal (Part 1)**
>
> Thank you for your review. Regarding your questions:
>
> > W1. The paper claims to have discovered the "trajectory invariance principle" for efficient hyperparameter tuning. However, there is one critical thing: the paper only conducts its experiments on the AdamW optimizer (and with the standard setting as well). Note that in AdamW, the weight decay is a simple, direct decay applied to the weights, separated from the gradient update. Therefore, I assume that the joint effect of learning rate and weight decay parameter might be simple, leading to a simple term ELR that can capture the trajectory (in some sense). However, this does not mean that the analysis can be applied to other optimizers, including Vanilla Adam, and especially LION, Muon, of which the update principles are very different.
>
> We present trajectory invariance w.r.t. ELR for other optimizers, including Muon (kimi), also known as Moonlight, Muon (spectral), and Lion. The difference between Muon (kimi) and Muon (spectral) is solely in the adjusted LR ratio.
>
> Results are shown in **Figure 25 in Appendix A.8**. We see that all optimizers in our experiments successfully exhibit trajectory invariance w.r.t. ELR in terms of loss curves.

---

> ### Author Response · Authors · 2025-11-21
> **Rebuttal (Part 2)**
>
> > W2. The above suggests that the finding might not be a "principle", but rather an "artifact" that happens in some sense only for AdamW. In such a case, I expect a (simplified) theoretical analysis on why this phenomenon happens for AdamW (or at least some strong intuitive explanation with concrete evidence), so that the paper might be interesting in some sense. Unfortunately, this is not the case for this paper.
>
> The experiments across various optimizers suggest that the principle we find is not an "artifact" specific to AdamW and is likely to be more general.
> We expect this invariance to apply to optimizers using decoupled weight decay. We do not claim that this principle applies to vanilla Adam.
>
> Below, we provide an intuitive explanation of the trajectory invariance phenomenon, in the case of a constant learning rate, for optimizers using decoupled weight decay. The update of optimizers with decoupled weight decay can be unified as:
> $$
>     w_{t+1} = (1-\eta\lambda)w_t + \eta\lambda (u_t/\lambda),
> $$
> where $u_t$ is the update given by different optimizers.
> Let $g_t$ be the stochastic gradient in the current iteration. For different optimizers, $u_t$ is defined as:
>
> - AdamW: $u_t = -\dfrac{m_t}{\sqrt{v_t} + \epsilon}$
> - SGD: $u_t = -g_t$
> - Muon: $u_t = -\text{NewtonSchulz}(m_t)$
> - Lion: $u_t = -\text{Sign}\bigl(\beta_1 m_{t-1} + (1-\beta_1) g_t\bigr)$
>
>
> Then, the whole process by which trajectory invariance emerges is:
> - The $\ell^2$ update norm $||u_t||$ stabilizes at the early stage of training, instead of diminishing to zero.
> - The $\ell^2$ parameter norm $||w_t||$ increases and then becomes stable, and this stable value is proportional to $\sqrt{\eta/\lambda}$.
> - When the $\ell^2$ parameter norm becomes stable, the effective learning rate (relative update size) is $\gamma = \eta/||w||$.
> - Substituting the stable parameter norm $||w|| \propto \sqrt{\eta/\lambda}$ gives the effective learning rate $\gamma \propto \sqrt{\lambda\eta}$. We hypothesize that the loss curves will behave similarly as long as the relative update size is invariant.
>
>
> **The stabilization of $\ell^2$ update norm $||u||$.**
>
> Earlier this year, [3] observed that AdamW’s update RMS norm is usually around 0.2 to 0.4. In our experiments, we also observe that AdamW’s update RMS norm stabilizes quickly (around 10,000 steps) to around 0.2.
> The stabilization of the update norm also happens for Muon and the Lion optimizer.
> A derivation for AdamW can be found in [4]. We will include it in the revised paper for completeness.
>
> **The stabilization of $\ell^2$ parameter norm $||w||$.**
>
> The EMA formulation is a good way to understand the behavior of optimizers using decoupled weight decay.
> It tells us that we only average over roughly the last $1/(\eta\lambda)$ minibatch updates to the current weight iterate [2]. Therefore, after the update norm stabilizes, the parameter norm can also become stable under the effect of EMA. Additionally, the speed of stabilization is controlled by the learning rate: a smaller learning rate leads to slower stabilization.
>
> The stable value is proportional to $\sqrt{\eta/\lambda}$.
> Some derivations of this stable value can be found in [5,6]. We will include it in the revised paper for completeness.
>
> **Deriving effective learning rate with stable parameter norm.**
>
> When the $\ell^2$ parameter norm becomes stable, we can view the update direction as living in the tangent space, i.e., orthogonal to the parameter direction, and rotating the parameters on a hypersphere. Therefore, the relative update, or effective learning rate, is naturally $\eta/|w_t|$. More specifically, we have
> $$
>     w_{t+1} = (1-\eta\lambda){||w_{t}||} \left(\frac{w_t}{{||w_{t}||}} + \frac{\eta}{{(1-\eta\lambda)||w_{t}||}} u_t\right).
> $$
> Using the approximation that ${||w_{t}||} \approx {||w_{t+1}||}$ and $1-\eta\lambda \approx 1$, we get
> $$
>     \frac{w_{t+1}}{||w_{t+1}||} \approx \frac{w_t}{{||w_{t}||}} + \frac{\eta}{{||w_{t}||}} u_t.
> $$
> In summary, when the parameter norm is stable, the effective learinng rate is $\eta/||w||$.

---

> ### Author Response · Authors · 2025-11-21
> **Rebuttal (Part 3)**
>
> > W3. Moreover, if we reluctantly accept the experiments restricted to AdamW, I still think that the experiments are insufficient. For example, the authors explicitly acknowledge that optimizer states of AdamW, like $\beta_2$, are "important" and "influence" the training. However, the authors only consider the experiments with one value of $\beta_2$
>
> We show trajectory invariance w.r.t. ELR for a different value of $\beta_2=0.995$ in AdamW in **Figure 26 (b) in Appendix A.8**. We see that AdamW with a different $\beta_2$ successfully exhibits trajectory invariance w.r.t. ELR in terms of loss curves.
>
> We would like to clarify that, although we state in the conclusion that "$\beta_2$ is important and influences training," we also write that "we expect the trajectory invariance between LR and WD to persist under different $\beta_2$ values." Our main intention with those sentences is to convey that incorporating $\beta_2$ into the invariance relation between LR and WD may be feasible, but the dependence is likely more complex than a simple effective learning rate given by the product of LR and WD.
>
> ---
>
>
>
> [1] Scaling Optimal LR Across Token Horizons.
>
> [2] How to set AdamW’s weight decay as you scale model and dataset size.
>
> [3] Muon is Scalable for LLM Training.
>
> [4] https://kexue.fm/archives/11267
>
> [5] https://kexue.fm/archives/11307
>
> [6] Rotational Equilibrium: How Weight Decay Balances Learning Across Neural Networks

---

> ### Author Response · Authors · 2025-11-28
> **Looking forward your feedback**
>
> Dear Reviewer J8na,
>
> We hope that our responses adequately address your concerns. As the deadline of this discussion phase is approaching, we warmly welcome further discussion regarding any additional concerns that you may have.
>
> Thank you for the time and appreciation that you have dedicated to our work.
>
> Best regards,
>
> Authors of submission 5475

---

> ### Comment · Reviewer_J8na · 2025-11-28
> **Official comments by Reviewer J8na**
>
> I thank the authors for the detailed comments. Sorry for the delay, I was trying to consider this rebuttal carefully.
>
> 1. I find the new experiments provided in Figures 25-27 make the paper a bit more reliable.
> 2. For the "theoretical justification", it is "persuasive heuristically". It is fine for me by now.
>
> Therefore, I will __increase my score to 6, but with confidence 2__. Good luck with the submission.

---

### Author Response · Authors · 2025-12-03
**A Summary of Rebuttal (Part 1)**

Dear AC,

We are deeply appreciative of the time and effort you have devoted to reviewing our paper. To assist you in forming a recommendation, we provide a concise summary of the author–reviewer discussion.

**Paper revision**: In the updated paper, we have made the following revisions:
- **Experiments on more optimizers and larger models.** In **Appendix A.8**, we provide experiments on trajectory invariance with respect to effective learning rate (ELR) with the Lion and Muon optimizers, with AdamW ($\beta_2=0.995$), and with a 1B language model. All the experimental results are consistent with our main figure, which indicates the universality of trajectory invariance.
- **Theory-inspired intuitive explanation.** In **Appendix C**, we provide an intuitive explanation of the trajectory invariance phenomenon from the viewpoint of parameter norm and relative update size. Our explanation applies to optimizers using decoupled weight decay, which is exactly justified by our experiments on AdamW, Muon, and Lion.
- **Discussion about previous scaling-law results.** In **Appendix A.6**, we discuss the main differences in the setup for fitting learning rate (LR) scaling laws between our work and previous studies, which we believe are the main reason for obtaining different results. In particular, we reproduce the decreasing scaling trend for learning rate at fixed weight decay, which justifies our explanation.
- **Implementation details.** In **Appendix B**, we discuss implementation details for computing gradient noise and population gradient norm, and for fitting scaling laws.


---

Next, regarding the reviewers, we provide a summary of the reviews and our replies. We believe our rebuttal has substantially addressed their main concerns.

1. **Reviewer J8na (score: 4; comment: will increase my score to 6, but with confidence 2)**
    - **Verification on more optimizers.** We show trajectory invariance for the Muon and Lion optimizers, effectively demonstrating the generality of this phenomenon.
    - **Principle or artifact?** On one hand, the experiments with more optimizers show that this is not an artifact; on the other hand, we provide a theory-inspired intuitive explanation for a more general class of optimizers with decoupled weight decay.
    - **Problems on experiment about single $\beta_2$ in AdamW.** We show trajectory invariance with another value $\beta_2 = 0.995$ in AdamW.

2. **Reviewer zBM1 (score: 2; not replied)**
    - **Writing problems.** We revise the corresponding figures and texts in the paper.
    - **Analytical discussion.** We provide a theory-inspired intuitive explanation for a general class of optimizers with decoupled weight decay. The simple explanation provided by the reviewer is not fully correct. In particular, the update does not tend to converge as training progresses. We believe our explanation is both more concrete and more persuasive.
    - **Justification for claims about scaling laws.** We clarify that we study and fit scaling laws with a broader range of hyperparameters, and that the results are not an artifact of specific conditions, such as a particular optimizer or hyperparameter choice. In fact, when we use the settings in previous studies, such as fixing weight decay, we can exactly reproduce their results (decreasing learning-rate scaling laws). We also compare the differences in the setup for fitting scaling laws between our work and previous work. Both the comparison and the reproduction are in Appendix A.6.
    - **The generality of the proposed invariance.** Our experiments on more optimizers (Lion, Muon), AdamW with other hyperparameters ($\beta_2 = 0.995$), and larger models effectively show the generality of the proposed invariance.
    - **Omitted implementation details.** We provide implementation details in Appendix B.

---

### Author Response · Authors · 2025-12-03
**A Summary of Rebuttal (Part 2)**

3. **Reviewer vF4o (score: 8; not replied)**
    - **Results on larger models and more optimizers.** We conduct experiments with more optimizers, including Muon and Lion, and 1B models, to effectively show the generality of the trajectory invariance phenomenon.
    - **Why different conclusions in scaling laws?** We attribute the differences in results to differences in setup, and compare the main differences in the setup for fitting LR scaling laws between our work and previous studies. We attribute the difference between decreasing and constant scaling trends to whether weight decay is fixed or not. We also suspect that the difference between decreasing and increasing trends is due to the choice of final learning rate (fixed ratio of max LR vs. fixed final LR).

4. **Reviewer qowG (score: 2; comment: maintain original score)**
    - **Limited model scale.** Both points made by the reviewer concern the limited scaling problem, which is also the **only** point raised by the reviewer.
    - On the one hand, we successfully show the trajectory invariance phenomenon in a 1B language model. On the other hand, we show experiments on more optimizers and provide an insightful theory-inspired explanation, which together provide more evidence of its generality. We think the more intrinsic aspect behind scaling is this generality. With these evidences from many perspectives, we believe we have already demonstrated the generality of trajectory invariance; we do not need a larger model scale to further show its generality. Additionally, we believe it still holds at larger model sizes, such as 7B, 30B, and 200B.
    - For the fitted scaling-law results, we focus entirely on scaling laws across dataset size. We note that in many works on hyperparameter scaling laws, scaling laws across dataset size are consistent across each independent model size. We also note that many prior studies conduct their main analyses at moderate model scales when the focus is not on model size itself, but on other aspects such as dataset size.
    - Finally, we also want to highlight our use of compute. Concretely, the total compute used in our experiments is at least 1.6e21 FLOPs, which corresponds to roughly 19.2K GPU hours. This is a substantial budget, and concentrating it on smaller models allows us to characterize the phenomenon — especially trajectory invariance to effective learning rate (ELR) — with much higher fidelity and confidence than if we had run a few sparse configurations at larger scales.

Overall, we think our research is insightful and facilitates our understanding of deep learning optimization, achieved with only a limited compute budget.

We hope this summary facilitates your review of our revised manuscript and rebuttal.

Best regards,

Authors of submission 5475

---

### Meta-Review · Area_Chair_8pv3 · 2025-12-26

**Summary:**

This submission proposes a “trajectory invariance” phenomenon in LLM pre-training dynamics, arguing that loss and related optimization statistics collapse along a salient direction that combines learning rate and weight decay (effective learning rate), and that the relevant invariance can shift across training regimes (e.g., iteration-limited vs. sufficient-iteration settings and across batch-size regimes). Reviewers generally found the empirical observation interesting and potentially useful as a practical heuristic to reduce the dimensionality of hyperparameter search. However, the discussion also surfaced significant disagreement about whether the current experimental scope and framing justify the paper’s stronger claims, especially those positioned as refinements or challenges to existing scaling-law conclusions. The rebuttal and revision added evidence (additional optimizers, a limited 1B-scale check, and an intuitive explanation plus implementation details), which improved perceived reliability for some reviewers, but did not resolve the core concern that the scaling-law narrative remains under-supported relative to the breadth of the claims. Given the remaining uncertainty around generality and the strength of the scaling-law statements, I recommend rejection.

**Reviewer Concerns:**

The central technical concern was generality beyond the initial (AdamW-centric) evidence and, separately, whether the paper’s scaling-law conclusions are warranted without broader cross-scale validation. The rebuttal partially addressed the “artifact vs. principle” critique by adding experiments with additional optimizers (e.g., Muon and Lion) and an additional AdamW setting, and by providing a heuristic explanation tied to decoupled weight decay; these steps increased confidence that the invariance is not purely an AdamW-specific coincidence. Nevertheless, concerns remain that the evidence for revising/challenging scaling-law viewpoints is still narrow in scope: the most critical reviewer feedback emphasized that robust scaling-law claims typically require systematic cross-scale comparisons, whereas the added 1B experiments are limited and do not constitute a comprehensive cross-scale study. In addition, reviewers raised questions about sensitivity to experimental choices (e.g., whether weight decay is tuned, schedule endpoints, and batch-size regime), which makes the paper’s strongest “refinement/challenge” language feel stronger than what is securely established. Presentation/clarity issues were also noted (definitions, figure readability, and completeness of appendices); while the revision appears to improve these aspects, they contributed to lower initial confidence and do not fully offset the remaining evidentiary gaps.

**Reviewer Scores:**

Reviewer vF4o is strongly positive (score 8) and did not provide any post-rebuttal comment indicating a change; based on the discussion, it is reasonable to assume they would remain at 8. Reviewer J8na provided an explicit post-rebuttal update stating they would raise their score from 4 to 6 (with low confidence), so I treat their inferred final stance as 6. Reviewer zBM1 did not participate in the discussion after the rebuttal and initially raised substantial concerns; absent contrary signals, I infer they would remain at 2. Reviewer qowG engaged after the rebuttal and explicitly stated that, despite appreciating the added experiments and intuition, the core concerns remain and they would maintain the original score; I therefore infer they would remain at 2.

---

### Decision · Program_Chairs · 2026-01-26

Reject